# *A Stitch in Time Saves Nine*: DETECTING AND MITIGATING HALLUCINATIONS OF LLMS BY ACTIVELY VALIDATING LOW-CONFIDENCE GENERATION

## ABSTRACT

Recently developed large language models (LLMs) have achieved remarkable success in generating fluent and coherent text. However, these models often tend to 'hallucinate' which critically hampers their reliability. In this work, we address this crucial problem and propose an approach that actively detects and mitigates factual hallucinations during the generation process. Specifically, we first identify the candidates of potential hallucination leveraging the model's *logit output values*, check their correctness through a *validation* procedure, mitigate the detected hallucinations via *prompting*, and then continue with the generation process. This active intervention also facilitates in preventing the propagation of hallucinations in the LLM's output. Through extensive experiments with GPT-3.5 (text-davinci-003) on the 'article generation task', we first demonstrate the individual efficacy of our detection and mitigation techniques. Specifically, we achieve a detection recall of $\sim 88\%$ and successfully mitigate $57.6\%$ of the correctly detected hallucinations. Importantly, our mitigation technique does not introduce new hallucinations even in the case of incorrectly detected hallucinations, i.e., false positives. Then, we show that the proposed active detection and mitigation approach successfully reduces GPT-3.5's hallucinations from $47.5\%$ to $14.5\%$. We further demonstrate the effectiveness and wide applicability of our approach through additional experiments with different types of questions (multi-hop and false premise) and with another LLM from a different model family (Vicuna). In summary, our work contributes to improving the reliability and trustworthiness of LLMs, a crucial step en route to enabling their widespread adoption in real-world applications.

## 1 INTRODUCTION

Hallucination in the context of language models refers to the generation of text or responses that seem syntactically sound, fluent, and natural but are factually incorrect, nonsensical, or unfaithful to the provided source input Maynez et al. (2020); Holtzman et al. (2020); Ji et al. (2023); Koehn & Knowles (2017). These hallucinations can lead to serious consequences such as spreading of misinformation and violation of privacy. This critically hampers models' reliability and limits their widespread adoption in real-world applications. Thus, **in this work, we focus on addressing the crucial problem pertaining to LLMs' hallucinations.**

We propose to actively 'detect' and 'mitigate' hallucinations during the generation process. This is crucial as we show that when a sentence generated by a model is hallucinated, that increases the chances of hallucination in the subsequently generated sentences of the model, i.e., hallucinations often propagates in the model's output. This can be attributed to the autoregressive nature of the LLMs and discrepancy between the training and inference time decoding. Specifically, during the training time, the model is encouraged to predict the next token conditioned on the ground-truth prefix sequences. However, at inference time, the model generates the next token conditioned on the historical sequences previously generated by itself. Thus, actively detecting and mitigating hallucinations during the generation process also facilitates in preventing the propagation of hallucinations in the subsequent generation. We divide our approach into two stages: **Detection** and **Mitigation**. Figure 1 illustrates the key steps of our approach. In order to address the complex task of detecting and mitigating hallucinations, we break it down into multiple simpler steps.

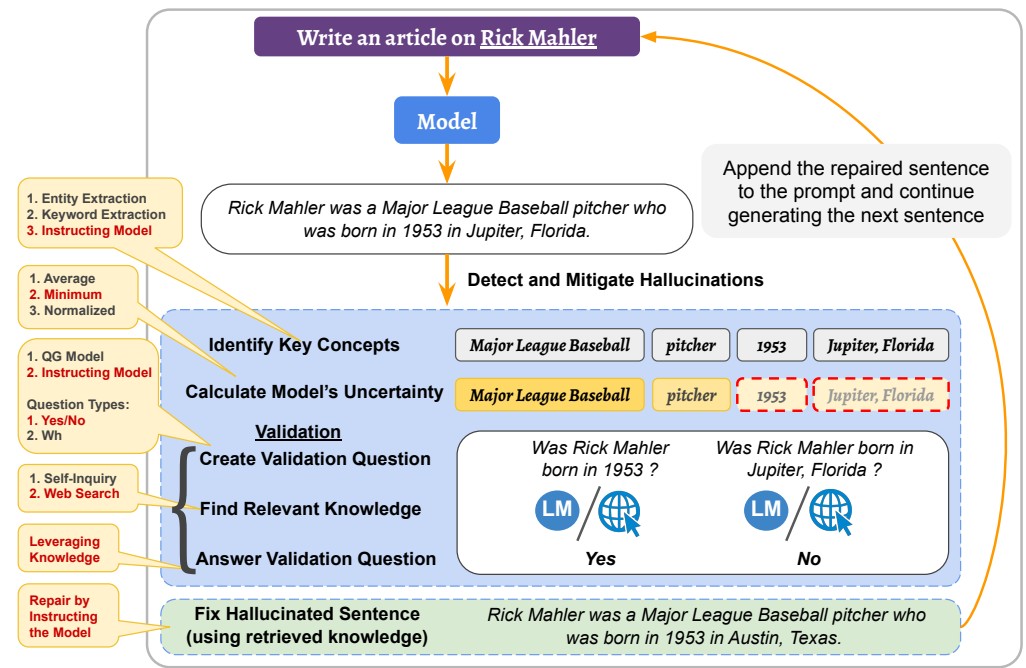

Figure 1: Illustration of the **proposed active detection and mitigation approach**. Different techniques for each step are mentioned on the left with the preferred one highlighted in red.

In the **hallucination detection** stage, we first identify the candidates of potential hallucination, i.e., the key 'concepts' of the generated sentence. Next, leveraging the logit output values of the model, we calculate model's 'uncertainty' on the identified concepts. We demonstrate that this uncertainty score provides a signal for hallucination. However, we note that this is an additional signal and not a necessary requirement for our approach. Then, we check the correctness of the 'uncertain' concepts through a *validation* procedure to detect hallucinations. This is followed by **hallucination mitigation** where we 'repair' the sentence via prompting using the retrieved knowledge as evidence. We conduct a systematic and wide study exploring multiple techniques for each step of the approach. Interestingly, we show that simply *instructing* the LLM does fairly well on these steps.

In our experimental setup, we prompt the model to write about specific topics from diverse domains such as Sports, Politics, Music, etc. Then, we annotate the correctness of the first five generated sentences for each topic. We first highlight the two findings that motivate our approach, i.e., the phenomenon of propagation of hallucination in the model's output and the utility of logit output values in detecting hallucinations. Then, we show the individual efficacy of our detection and mitigation techniques. Specifically, we achieve a detection recall of $\sim 88\%$ and successfully mitigate $57.6\%$ of the correctly detected hallucinations. Importantly, our

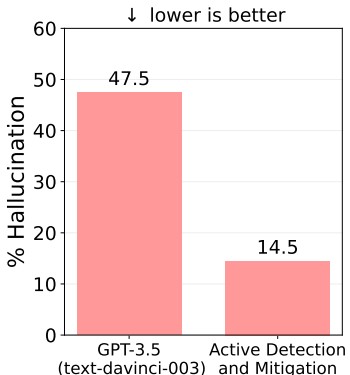

Figure 2: Comparing % hallucination in the output of GPT-3.5 with our active detection and mitigation approach on 'article generation task'.

mitigation technique does not introduce new hallucinations even in the case of incorrectly detected hallucinations, i.e., false positives. Then, we show that the proposed active detection and mitigation approach successfully reduces GPT-3.5 (text-davinci-003) hallucinations from $47.5\%$ to $14.5\%$ (Figure 2). To demonstrate the effectiveness and wide applicability of our approach in addressing hallucinations, we further present three additional studies: (1) Efficacy with another LLM (Vicuna fine-tuned on LLaMA-2) from a different model family, (2) Adapting the approach to answer Multi-hop questions, and (3) Assessing it on False premise questions.

## 2 APPROACH

As motivated in Section 1, we propose to iteratively generate sentences and actively detect and mitigate hallucinations during the generation process. This is crucial to prevent the propagation of hallucination in the model's output. As shown in Figure 1, we break down the complex task into **detection** and **mitigation** stages which are further broken down into simpler steps to achieve better performance. In Section 2.1, we detail the hallucination detection steps, viz., identifying the candidates of potential hallucination (2.1.1), calculating model's uncertainty on the concepts (2.1.2), and checking the correctness by creating validation query (2.1.3), finding relevant knowledge (2.1.4), and verifying the information (2.1.5). We describe various techniques to achieve the objective of each step and **indicate the preferred technique with (*)**. In Section 2.2, we detail our mitigation approach where we 'repair' the hallucinated sentence by removing/substituting the hallucinated information and incorporating the correct information from the retrieved knowledge. We can also utilize this knowledge as context for subsequent generation. Table 3 shows the instructional prompts and Appendix B provides all the supporting details of all steps of the approach.

### 2.1 HALLUCINATION DETECTION

### 2.1.1 IDENTIFY KEY CONCEPTS

We start by identifying the candidates of potential hallucination, i.e., the important concepts from the generated sentence. We identify these concepts because validating the correctness of the entire sentence at once is infeasible as a sentence often contains multiple different facets all of which can not be validated at once. In contrast, individually validating correctness corresponding to the concepts provides opportunities for accurately detecting hallucinations. Note that a concept is essentially a span of text consisting of one or more tokens. We study the following techniques for this step:

**Entity Extraction:** Entities are typically important parts of a sentence, thus, we explore using an off-the-shelf entity extraction model to identify the concepts. A limitation of this method is that a concept need not necessarily be an entity.

**Keyword Extraction:** Addressing the limitation of the entity extraction method and additionally identify the non-entity concepts, we explore using an off-the-shelf keyword extraction model.

**\*Instructing the Model\*:** Since state-of-the-art LLMs perform remarkably well on a wide range of tasks, in this technique, we directly instruct the model to identify the important concepts from the generated sentence. We specify the instructional prompt in Table 3 and further details in B.1.

Table 4 (B.1) illustrates examples of concepts identified using the three techniques. It shows that the entity extraction model often misses many important concepts while the keyword extraction model identifies a number of insignificant concepts also. In contrast, the instruction technique successfully identifies all the important concepts. Moreover, it doesn't require calling a task-specific tool (entity or keyword extraction model). Thus, we regard it as our preferred technique for this step.

### 2.1.2 CALCULATE MODEL'S UNCERTAINTY

GPT-3 and several other SOTA models also provide logit values in their output. Thus, we study if these values can be utilized to detect hallucinations. Consider a concept consisting of $n$ tokens and having the maximum softmax probabilities as $p_1, p_2, p_3, ..., p_n$ for the $n$ token positions respectively. We study three different techniques for calculating a **probability score** for a concept:

**Average of Probabilities** (AVG $[p_1, p_2, ..., p_n]$) , **Normalized Product of Probabilities** ($[p_1 \times p_2 \times ... \times p_n]^{1/n}$) and , **\*Minimum of Probabilites\*** (MIN $[p_1, p_2, ..., p_n]$) . Here, 'MIN' is our preferred technique as the others may average out the effect of model's uncertainty on the tokens while low probability in even one token of a concept provides sufficient evidence of the model being uncertain in its generation. For e.g., if the model is uncertain about name of the USA president then its uncertainty on the first token ('Joe') would be high but on the next token ('Biden') would be very low as token 'Joe' is frequently followed by token 'Biden' in raw text. Thus, Averaging or Normalizing the probabilities will have a limited capability to capture this signal in comparison to Minimum.

In 3.1.2, we show that this score (especially 'MIN') indeed provides a signal for hallucination, i.e., **the more uncertain the model is on a concept (low probability score), the more likely it is to**

**be hallucinating about that concept**. Figure 13 compares the performance of the three probability calculation techniques. Thus, we utilize this signal and check for hallucinations with respect to the uncertain concepts using our validation procedure (2.1.3-2.1.5).

**In the absence of logit output values** (for models that do not provide these values in their prediction response), all or some heuristically selected concepts (depending on the computational and latency budget of the system) can be passed to the validation stage for detecting hallucinations.

### 2.1.3 CREATE VALIDATION QUESTION

Our validation procedure for a concept starts with creation of a question that tests the correctness of the information (in the generated sentence) pertaining to the concept. We study creating **Yes/No Questions** as illustrated in Table 5 using **Question Generation Tool** and *Instructing the Model*.

In instruction technique, we directly prompt the model to create a validation question checking the correctness of the information about the selected concept. Similar to the concept identification step, it is our preferred technique as it does not require calling a task-specific tool. We note that instead of Yes/No questions, **Wh-questions** can also be used for validation. We prefer Yes/No questions as it is relatively easier to verify their answers. We explore Wh-questions for a study in Section 4.2.

### 2.1.4 FIND RELEVANT KNOWLEDGE

We explore two ways of retrieving the relevant knowledge to answer the validation question.

**\*Web Search\*:** Web search provides several benefits such as generality, wide coverage, and information freshness. We use Bing search API for retrieving the knowledge. However, we note that any other search API or knowledge corpus can also be utilized for this purpose.

**Self-Inquiry:** Here, we leverage the parametric knowledge of the LLM and directly prompt it to answer the validation question. Though it does not require external knowledge, it has drawbacks such as lack of a reliable strategy to extract the parametric knowledge and knowledge staleness.

Note that our proposed approach has several benefits pertaining to retrieval: (a) it does not retrieve knowledge when it is not required, i.e., when the model is already sufficiently confident (since we show that it is less likely to hallucinate in such scenarios), (b) it individually retrieves knowledge pertinent to the concept(s) on which the calculated probability score is low thus providing it sufficient and relevant context for accurate validation and mitigation (Section 2.2).

### 2.1.5 ANSWER VALIDATION QUESTION

Now, we prompt the model to answer the validation question leveraging the retrieved knowledge as context and verify its response. If the validation procedure succeeds for all the uncertain concepts then we continue generating the next sentence; otherwise, we interrupt the generation and mitigate the potential hallucination in the sentence before continuing the subsequent generation.

**Order of Validation of Concepts:** Validation of different concepts can be done in a sequence (in ascending order of their calculated probability score) or in parallel. However, running this in parallel would require starting multiple threads which may not be supported by all machines. Thus, in this work, we study only the sequential validation strategy but note that it can be made more efficient by running it in parallel. We regard this sequential validation as a greedy exiting strategy as we proceed to the mitigation stage on detection of the first hallucinated concept.

### 2.2 HALLUCINATION MITIGATION

For mitigating hallucination in the generated sentence, we instruct the model to repair the generated sentence by removing/substituting the hallucinated information and incorporating the correct information using the retrieved knowledge as evidence (Table 3 shows the instructional prompt).

We note that the result of our validation procedure is contingent on the retrieved knowledge and the model's ability to leverage that knowledge in answering the validation question. In Section 3.2, we show that our approach performs well on this task and achieves a high recall demonstrating its efficacy at detecting hallucinations. Moreover, we show that our mitigation approach does not introduce new hallucinations even in the case of incorrectly detected hallucinations (false positives).

In B.3, we elaborate on our design decisions such as **benefit of "active" approach over the "posthoc" approach**, **impact on inference cost**, and **justification of procedural methodology**.

# 3 EXPERIMENTS AND RESULTS

We first highlight the two findings that motivate our approach (in 3.1.1 and 3.1.2). Then, we show the individual efficacy of our detection and mitigation techniques in 3.2. Finally, in 3.3, we show the effectiveness of our proposed active detection and mitigation approach.

**Data and Annotation:** In our experimental setup, we prompt the LLM to write about a given topic. We use topics from diverse domains as shown in Figure 3. In each domain, we include different kinds of topics; for instance, Sports includes sportspersons, teams, and games; Music includes musicians, songs, music labels, and bands; Politics includes politicians, political parties, and elections, etc. We use a total of 150 topics in our data. For selecting the names of people, we randomly sample from the top 20% of longest articles in WikiBio dataset Lebret et al. (2016) as done in Manakul et al. (2023). Similarly, we sample from the longest Wikipedia articles for the other topics. This is done to avoid obscure or ambiguous topics.

For each topic, we give the following input prompt to the model (text-davinci-003 and Vicuna-13B): 'Write an article about <topic>'. Then, we (the authors) annotate the correctness of the first five sentences generated by the model. For this annotation, we look at search results from the web to find the relevant knowledge that either supports or contradicts the information in the sentence. In some cases, multiple web searches were required to check the correctness of dif-

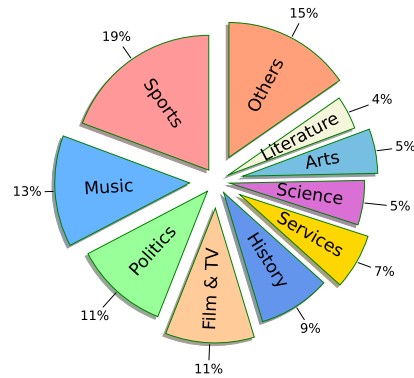

Figure 3: Distribution of instances across different domains in our topic set.

ferent facets of a sentence. Furthermore, in a small number of cases, we could not find information supporting or contradicting the information in the generated sentence, we marked it as a case of extrinsic hallucination. We opt for this **expert annotation strategy** because despite the annotation task being a simple binary classification task, it requires considerable effort to check the correctness which can not reliably be collected via crowdsourcing. In addition to the **sentence-level annotation**, we also annotate correctness at **concept-level** (detailed in 3.1.2). We will release both sentence and concept-level annotations to facilitate a systematic future research in this direction.

## 3.1 MOTIVATING FINDINGS

### 3.1.1 PROPAGATION OF HALLUCINATION IN THE MODEL'S OUTPUT

Since we consider five sequentially generated sentences generated by the model for each topic, we investigate the relationship between '*hallucination in a generated sentence*' and '*hallucination in any previously generated sentences*' for an input. Since there are two binary variables, there exist four possibilities in this relationship, represented by YY, NY, YN, and NN in Figure 4. The figure demonstrates this relationship for sentences 2, 3, 4, and 5 (since no previously generated sentence for sentence 1) aggregated over all the topics in our dataset. Observations are as follows:

(a) **YY > NY**: Cases YY and NY correspond to the scenario when there is a previous hallucination. It can be observed that YY is considerably greater than NY implying that *when there is hallucination in the previously generated sentences, a sentence is more often hallucinated*.

(b) **YY > YN**: In YY and YN, the generated sentence is hallucinated. Here, YY is greater than YN implying that *a generated sentence is hallucinated more when there is hallucination in the previously generated sentences as compared to when there is no hallucination in the previously generated sentences*.

(c) **NN > YN**: *When there is no hallucination in the previously generated sentences, a generated sentence is more likely to be correct, i.e., it is less often hallucinated*.

(d) **NN** > **NY**: *A generated sentence is 'correct' more when there is no previous hallucination as compared to when there is a previous hallucination.*

This shows that hallucination in a sentence increases the chances of hallucinations in the subsequently generated sentences, i.e., hallucination often propagates and thus **actively detecting and mitigating them can fix the current hallucination and also prevent its propagation in the model's output**.

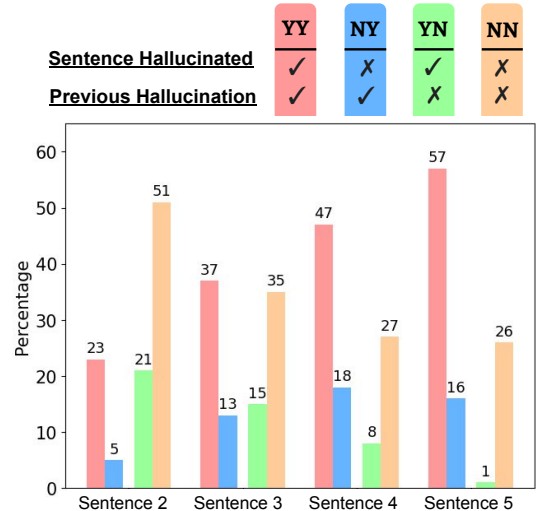

### 3.1.2 LOGIT OUTPUT VALUES PROVIDE A SIGNAL FOR HALLUCINATION

To study the relationship between logit values and hallucination, we annotate correctness at concept-level also (in addition to sentence-level annotations described earlier). Specifically, for each identified concept, we mark whether the information about it in the generated sentence is hallucinated or not. Table 7 shows examples of both sentence and concept-level annotations. Figure 5 shows the trend of hallucination with our calculated probability scores. For sentence-level, we use the minimum across tokens of all its identified concepts as the probability score, and for concept-level, we use the minimum across the concept's tokens as the probability score. The figure shows that **as the probability score increases (or uncertainty decreases), the tendency to hallucinate decreases.** This shows that the probability values can be utilized as a signal for hallucination, i.e., the low probability concepts can be considered as candidates of potential hallucination and their correctness in the sentence can be validated for detecting hallucinations.

Figure 4: Demonstrating relationship between 'hallucination in a generated sentence' and 'hallucination in previously generated sentences'. Bars YY, NY, YN, and NN correspond to four possibilities.

We compare efficacy of different probability calculation techniques at detecting hallucinations (in Appendix G) and show that the 'Minimum' technique achieves the highest area under the Precision-Recall curve. In Appendix G, we also demonstrate the benefit of identifying concepts for the detection task and show that the probabilities of the concept tokens provide a stronger signal for hallucination as compared to all tokens.

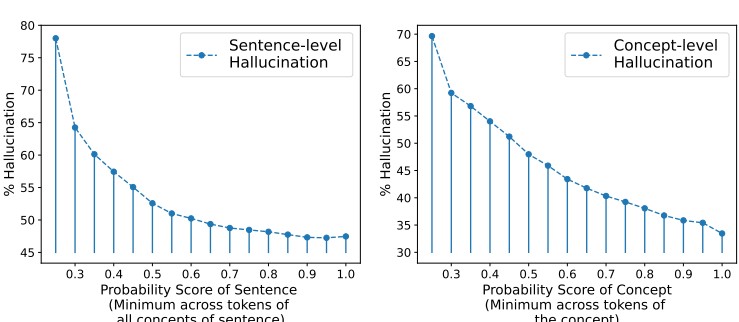

Figure 5: Trend of hallucination with the calculated probability score (Minimum technique) at both the sentence and concept levels. As the score increases, the tendency to hallucinate decreases.

## 3.2 HALLUCINATION DETECTION AND MITIGATION PERFORMANCE

**Detection:** In Table 1a and 1b, we compare the detection performance of self-inquiry and web search techniques at both sentence and concept-levels. For sentence-level results, we predict the sentence to be hallucinated if the validation procedure fails for any identified concept. Note that in these results, we do not leverage the uncertainty score to select concepts for validation, instead we validate all the identified concepts. We study the relationship of recall with probability thresholds in

Figure 11 (Appendix). The tables show that **web-search technique achieves considerably high recall and precision** in detecting the hallucinations. Here, we emphasize on the high 'recall' as we show that our mitigation approach does not introduce new hallucinations even in the case of incorrectly detected hallucinations, i.e., false positives.

**Mitigation:** On sentences where our validation procedure (using Web search) reports hallucinations, we apply our mitigation technique. We note that a sentence that is reported as hallucination can either be actually hallucinated (true positive) or not hallucinated (false positive). Table 2 shows the result of our method. It successfully mitigates the hallucination on 57.6% of the correctly detected hallucinations (True Positives). Furthermore, it achieves this at minimal 'deterioration' (3.06%), i.e., it incorrectly converts a minimal 3.06% of the non-hallucinated instances to sentences having incorrect information (hallucinated).

#### (a) Sentence level

| Technique | Accuracy | Hallucinated | | Not Hallucinated | |
|---|---|---|---|---|---|
| | | Prec. | Rec. | Prec. | Rec. |
| **Self-Inquiry** | 0.62 | 59.89 | 63.76 | 65.23 | 61.42 |
| **Web-Search** | 0.681 | 61.82 | **85.96** | 80.39 | 52.03 |

#### (b) Concept level

| Technique | Accuracy | Hallucinated | | Not Hallucinated | |
|---|---|---|---|---|---|
| | | Prec. | Rec. | Prec. | Rec. |
| **Self-Inquiry** | 0.65 | 47.96 | 45.85 | 73.37 | 74.98 |
| **Web-Search** | 0.75 | 58.17 | **87.68** | 91.69 | 68.30 |

Table 1: **Hallucination detection performance** of self-inquiry and web-search techniques. It also shows separate precision and recall on both hallucinated and non-hallucinated instances.

**Analyzing Mitigation Failures:** Table 8 and 9 (in Appendix) show examples where our mitigation technique successfully mitigates and fails to mitigate the hallucinations, respectively. We observe that in many of the failure cases, our technique fixes some hallucinated content of the sentences but fails to fix ALL the hallucinated content from them. Examples 1 and 2 in Table 9 correspond to this type of failure. Furthermore, in some of the failure cases, our technique results in a sentence that is no longer hallucinated but is not completely related to the topic. For instance, the fourth example in Table 9 about the topic 'Harry S. Kennedy'; the model generates "*Harry S. Kennedy was ... 35th President ...*" which is wrong and our mitigation technique modifies it to "*John F. Kennedy was ...*" which is factually correct but not related to the topic 'Harry S. Kennedy'. This happens because the output of the mitigation step is contingent on the information in the retrieved knowledge. We present further analysis in Appendix.

| Is Hallucinated? | | Percentage |
|---|---|---|
| **Before** | **After** | |
| ✓ | ✗ | 40.81% |
| ✓ | ✓ | 30.04% |
| ✗ | ✗ | 28.26% |
| ✗ | ✓ | 0.89% |

Table 2: Results after modifying the reported hallucinations.

### 3.3 ACTIVE DETECTION AND MITIGATION PERFORMANCE

The two findings in 3.1 motivate our approach in which we actively detect hallucinations leveraging the logit values and mitigate them during the generation process which further helps in preventing their propagation. Specifically, we iteratively generate sentences and when our detection method reports hallucination (by validating uncertain concepts), we repair the sentence using our mitigation method and then continue generating the next sentence. We demonstrated separate detection and mitigation efficacy in 3.2. Figure 2 compares the hallucination percentage in GPT-3.5's output and our "active" approach. It reduces the hallucination percentage from 47.4% to 14.53% which proves that the active intervention indeed successfully prevents hallucination propagation. In Figure 10 (Appendix), we plot this comparison for different categories of hallucinations and show that our approach does well in all the categories. We further elaborate on it in Appendix D.

## 4 ADDITIONAL EXPERIMENTS

To further demonstrate our approach's wide applicability, we present three additional studies and discuss other usecases in K. We present main results here and a detailed description in the Appendix.

### 4.1 EFFICACY WITH ANOTHER LLM (VICUNA-13B FINE-TUNED ON LLAMA-2)

Figure 6 compares hallucination % in the output of Vicuna-13B (on the 'article generation task') and with our proposed active detection and mitigation approach. We select Vicuna (v1.5) because

it is the SOTA open-source model. Our approach considerably reduces the hallucinations similar to the case with GPT-3.5 model. This study is conducted on 10 randomly sampled topics (i.e., 50 generated sentences) from the topic set described in Section 3. We note that similar to the setup with GPT-3.5 where we used instructional prompts with GPT-3.5 itself for all the steps of the approach (i.e., identifying key concepts, creating validation questions, etc.), following the same, here we use Vicuna-13B for all those steps. This result demonstrates **generality and applicability of our approach in reducing hallucinations of LLMs**.

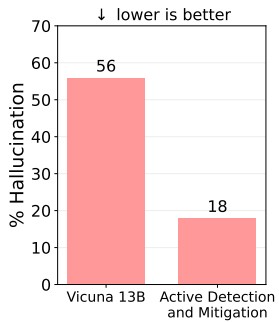

### 4.2 ADAPTING THE APPROACH FOR MULTI-HOP QUESTIONS

Here, we show that our approach can be adapted to improve the performance on multi-hop bridge questions (Table 10). Recall that our approach works by mitigating hallucination/incorrectness in the sentences generated by the model. Thus, if we can enable the model to answer these multi-hop questions step by step, then our active detection and mitigation approach can be applied to these steps, leading to correct predictions. To this end, we prompt the model and provide in-context examples demonstrating it to answer a given multi-hop question step by step. Appendix H shows the corresponding prompt

Figure 6: Comparing hallucination % for Vicuna-13B and our proposed approach on the 'article generation task'.

used for this purpose. Specifically, for a test question, the model generates the answer in multiple steps (one step at a time) and for each step, we apply our technique in which we first identify the low probability concepts from the sentence, validate their correctness using web search results, mitigate the hallucination (if detected), and then proceed to generate the next step. In our case study, we sample 50 multi-hop bridge questions from the validation set of HotpotQA Yang et al. (2018).

**Main Result (Figure 7):** First, we show the performance of GPT-3.5 which answers $54\%$ of the questions incorrectly. GPT-3.5 with in-context examples results in a slight improvement over the zero-shot performance. GPT-3.5 leveraging the knowledge retrieved from the web (using the question as search query) as context improves the performance and results in fewer incorrect predictions. Finally, we show the performance of our active detection and mitigation approach which results in considerably fewer hallucinations (just 26%), i.e., a higher percentage of correct answers. Table 11 (Appendix H) shows examples of responses generated using our approach. This demonstrates **our approach's effectiveness in improving performance on multi-hop QA**.

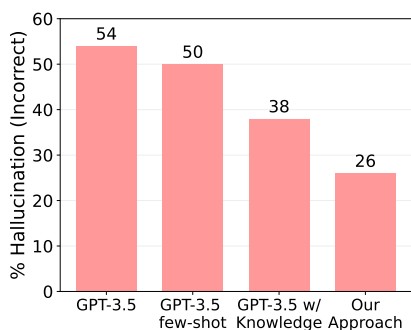

Figure 7: Comparing % hallucination on Multi-hop bridge Questions.

### 4.3 FALSE PREMISE QUESTIONS

LLMs perform remarkably well on a wide range of questions that are factually correct and make the right assumptions Khashabi et al. (2020); Brown et al. (2020); Zhang et al. (2022). However, users in real-world applications often ask questions that are based on false premises such as "Why energy is absorbed in exothermic reactions?" and "Why do floppy disks have higher storage capacity than USB drives?". We observe that SOTA models often struggle to appropriately respond to such questions; thus, they serve as another challenging evaluation setting. To this end, we conduct a case study and compile a set of 50 such adversarial questions, i.e., questions for which GPT-3.5 generates an incorrect response. Furthermore, we also create a true premise question corresponding to each false premise question (examples in Table 12).

**Approach:** An ideal response to such questions is application dependent; some applications may require identifying such questions and then abstaining on them like the selective prediction systems Kamath et al. (2020); Xin et al. (2021) while some applications may also require suggesting a 'rectified' question and providing response to that rectified question like the search engines. Our approach supports these requirements by using the validation and mitigation step on the given question.

Specifically, we first retrieve the relevant knowledge (via Bing Search using the question as query). Then, we apply our validation and mitigation technique, i.e., conditioned on the retrieved knowledge, we prompt the model to respond 'Yes' if the question makes factually correct assumptions, otherwise respond 'No'. If the response to this prompt is No, then we proceed to modify the question using the mitigation step. Table 13 shows the corresponding instructional prompts. This step enables identifying false premise questions and rectifying them to facilitate the system in providing an appropriate response. Importantly, we also show that our approach does not incorrectly modify a true premise question. This is crucial because if the user's question is correct then the system's response must be pertinent to that.

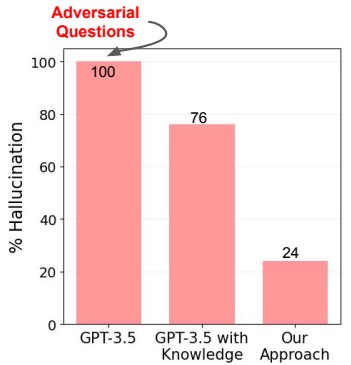

Figure 8: Comparing % hallucination on 'False Premise Questions'.

**Main Result (Figure 8):** As mentioned above, the questions in our evaluation set are adversarially collected, i.e., GPT-3.5 gives incorrect response to all of them. We evaluate the performance of GPT-3.5 when retrieved knowledge (via bing search) is given as additional context. We find that even with the knowledge, it manages to answer only 24% false premise questions correctly, i.e., hallucinates on the remaining 76%. In contrast, **our approach answers 76% questions correctly and hallucinates only on 24%.** Furthermore, we note that even in some of these 24% hallucinated responses, some of the individual sentences in the responses are correct. However, since we focus on complete answer correctness, we consider them as incorrect. Table 15 shows examples of responses on false premise questions generated by the GPT-3.5, GPT-3.5 with retrieved knowledge, and our active detection and mitigation approach.

## 5 RELATED WORK

One thread of research pertaining to hallucinations has focused on studying different causes of this phenomenon such as training data quality Wang (2019); Lee et al. (2022a), source-target divergence Dhingra et al. (2019), ill-suited modeling Aralikatte et al. (2021); Feng et al. (2020); Li et al. (2018), and stochasticity during inference Dziri et al. (2021); Tian et al. (2019); Lee et al. (2022b).

The other thread focuses on addressing this problem Manakul et al. (2023); Azaria & Mitchell (2023); Lee et al. (2022b); Du et al. (2023); Zhang et al. (2023). Manakul et al. (2023) propose to first sample multiple responses from the model and then measure the information consistency between them to detect hallucinations. They posit that when a model knows a given concept well, the sampled responses are likely to contain consistent facts. Another recent work Azaria & Mitchell (2023) trains a separate classifier that takes the LLM's activation values as input and predicts its truthfulness. Lee et al. (2022b) hypothesize that the randomness of sampling is more harmful to factuality when it is used to generate the latter part of a sentence than the beginning and propose factual-nucleus sampling that dynamically adapts the 'nucleus' p along the generation of each sentence. Du et al. (2023) propose an approach motivated by *The Society of Mind* and *multi-agent settings* in which multiple models individually propose and jointly debate their responses and reasoning processes to arrive at a common answer. We present an extended and detailed related work in Appendix A.

In our approach, we propose to actively detect and mitigate hallucinations by breaking down the complex task into multiple simpler steps. We utilize the logit values to identify candidates of potential hallucination, web search to validate the information and prompting to fix the hallucination.

## 6 CONCLUSION

In this work, we proposed an approach called active detection and mitigation to address the problem pertaining to the factual hallucinations of large language models. Through systematic and extensive experiments on several tasks such as article generation, multi-hop QA, and false premise QA, we showed that our approach considerably reduces the hallucinations of LLMs. Overall, by addressing the hallucination problem, our work contributes to improving LLMs' reliability and trustworthiness, a crucial step en route to enabling their widespread adoption in real-world applications.

ETHICS STATEMENT

The proposed approach considerably reduces the hallucination in the output of LLMs; however, it does not eliminate it completely. In other words, it certainly improves the correctness and reliability of LLMs but does not empower them with absolute correctness. We have provided a detailed description of the dataset in Section 3 which does not involve any kind of bias to the best of our knowledge. We will release both sentence and concept-level hallucination annotations to facilitate a systematic future research in this important research direction.

REPRODUCIBILITY STATEMENT

We note that the experiments conducted in this work are easy to reproduce. We have detailed our approach in Section 2 and provided all the additional information in Appendix B. All the experimental details are provided in Section 3 and Appendix C.

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

## APPENDIX

## A  EXTENDED RELATED AND CONCURRENT WORK

Advancements in the field of natural language processing have led to the development of models that possess an impressive ability to generate fluent and coherent text. However, these models are vulnerable to hallucinate in their output. Prior work Maynez et al. (2020); Huang et al. (2021); Ji et al. (2023) has categorized text hallucinations into two classes: Intrinsic (when the generated output contradicts the source content) and Extrinsic (when the generated output cannot be verified from the source content, i.e., it that can neither be supported nor contradicted by the source).

One thread of research pertaining to hallucinations has focused on studying different causes of this phenomenon such as **training data quality** Wang (2019); Lee et al. (2022a), **source-target divergence** Dhingra et al. (2019) (when a model is trained on noisy data with source-reference divergence, it may learn to generate text that is not necessarily grounded or faithful to the given source),

**ill-suited modeling** Aralikatte et al. (2021); Feng et al. (2020); Li et al. (2018), **stochasticity during inference** Dziri et al. (2021); Tian et al. (2019); Lee et al. (2022b) (decoding strategies that improve the generation diversity, such as top-k sampling, top-p, and temperature parameters, often result in increased hallucinations which could be attributed to the introduction of "randomness/stochasticity" while selecting tokens (from top-k or top-p) instead of choosing the most probable token while decoding), and **parametric knowledge bias** Longpre et al. (2021); Zhou et al. (2023b); Michel et al. (2019) in which Models often tend to prioritize the parametric knowledge (knowledge acquired during pre-training and implicitly stored in the parameters of the model) over the provided contextual knowledge resulting in hallucinations.

The other thread focuses on addressing the hallucination problem Manakul et al. (2023); Azaria & Mitchell (2023); Lee et al. (2022b); Du et al. (2023); Zhang et al. (2023). A recent work Manakul et al. (2023) propose a sampling-based hallucination detection approach in which they first sample multiple responses from the model and then measure the information consistency between the different responses. They posit that when a language model knows a given concept well, the sampled responses are likely to be similar and contain consistent facts; on the other hand, for hallucinated facts, stochastically sampled responses are likely to diverge and may completely contradict one another.

Another recent work Azaria & Mitchell (2023) leverages LLM's internal state to identify the truthfulness of a statement. Using an annotated dataset, they train a separate classifier that takes the LLM's activation values as input and predicts its truthfulness. Kadavath et al. (2022) have shown the utility of model's uncertainty values in detecting incorrectness in the model's responses by demonstrating that larger models are well-calibrated on multiple-choice and true/false questions. Lee et al. (2022b) hypothesize that the randomness of sampling is more harmful to factuality when it is used to generate the latter part of a sentence than the beginning of a sentence and propose a new sampling algorithm named factual-nucleus sampling that dynamically adapts the 'nucleus' p along the generation of each sentence.

Du et al. (2023) propose an approach motivated by *The Society of Mind* and *multi-agent settings* in which multiple models individually propose and jointly debate their responses and reasoning processes to arrive at a common answer.

Similar to our approach, concurrent work Gou et al. (2023); Chen et al. (2023); Zhao et al. (2023); Chern et al. (2023) also proposes to use external knowledge/tools to address the hallucination problem of LLMs. Other concurrent work FactScore Min et al. (2023) presents an evaluation method that breaks the model's generation into a series of atomic facts and computes the percentage of atomic facts supported by a reliable knowledge source. This supports the utility and effectiveness of our concept validation step. Though, it has considerable differences with our approach. Firstly, we validate the correctness of only the uncertain concepts which we identify using the logit output values. This is because we have shown that models tend to hallucinate more on these uncertain concepts. Secondly, we create a validation query pertinent to an uncertain concept and retrieve the pertinent information using that as the search query. Also, we not only detect the hallucinations but also repair them and then continue generating the next sentences. Different from our post-hoc approach that utilizes the pretrained LLM, Chen et al. (2023) finetunes a T5-large model as compact editor to denoise the corruptions to detect incorrectness in a given sentence. Another concurrent work Jiang et al. (2023) proposes active retrieval augmented generation. Our work differs from this in the following aspects. Firstly, we calculate uncertainty at a concept level (after identifying the important concepts using the LLM itself); in contrast, Jiang et al. (2023) actively trigger retrieval if any token of the sentence has a probability lower than a threshold. In this work (Appendix G), we have shown the importance of identifying the concept tokens in detecting hallucinations. This also ensures that the validation queries are created for the entire concept and not just some tokens. Furthermore, in this work, we demonstrate the necessity of active intervention using our novel propagation of hallucination study. Also, we demonstrate the effectiveness of our approach in multiple differnt settings including open-ended reference free text generation.

In summary, in our approach, we propose to actively detect and mitigate hallucinations by breaking down the complex task into multiple simpler steps. We utilize the logit output values (uncertainty) to identify candidates of potential hallucination (at concept-level), web search to validate the information, and prompting to fix the hallucination. We demonstrate the effectiveness and wide applicability

| Step | Prompt |
|------|--------|
| Input Prompt | `Write an article about {topic}` |
| Identify Important Concepts | `Identify all the important keyphrases from the above sentence and return a comma separated list.` |
| Create Validation Question | `For the above sentence about {topic}, generate a yes/no question that tests the correctness of {concept}.` |
| Answer Validation Question | `{search results} Answer the below question about topic in Yes or No based on the above context. {validation question}.` |
| Repair Hallucinated Sentence | `The above sentence has information that can not be verified from the provided evidence, repair that incorrect information and create a new sentence based on the provided evidence.` |

Table 3: Instructional Prompts corresponding to different steps of our approach.

of our approach on a variety of tasks, including article generation, multi-hop question answering, and false premise question answering.

## B  ADDITIONAL DETAILS OF THE APPROACH

In this section, we provide additional details of our approach. Table 3 shows the instructional prompts used for different steps of the approach. We note that the instruction technique is the preferred technique as it does not require calling a task-specific tool to achieve the corresponding objectives of the steps.

### B.1  IDENTIFY KEY CONCEPTS STEP

For keyword extraction, we explore a model[1] that uses Keyphrase Boundary Infilling with Replacement (KBIR) as its base model and is fine-tuned on the KPCrowd dataset Kulkarni et al. (2021).

Table 4 shows examples of concepts identified using the three methods, i.e., Entity Extraction, Keyword Extraction, and Instructing the Model. It shows that the entity extraction model misses many important concepts while the keyword extraction model identifies a lot of insignificant concepts also. In contract, instruction technique successfully identifies all the important concepts.

### B.2  CREATE VALIDATION QUESTION STEP

Table 5 shows examples of validation questions corresponding to each concept created via the instruction technique. It shows examples of both the question types, i.e., Yes/No and Wh questions. We prefer Yes/No questions as it is relatively easier to verify the answer of these questions.

We have also conducted evaluations of the efficacy of the instructions. Specifically, for the concept identification step, we studied randomly sampled 50 sentences. The instruction technique identified 155 concepts in total. It missed only 2 concepts (that too these missed concepts can only be loosely regarded as important in the context of the sentence). Furthermore, the efficacy of the validation and mitigation instructions is presented in Table 1 and 2, respectively.

We note that the overall efficacy of these techniques (and how well they serve their purpose) is evaluated by the overall improvement in reducing the hallucinations.

We also note that the LLM can be prompted in a different way also to achieve the same objective; however, the purpose of this work is to show that the complex task of addressing hallucinations in an end-to-end manner can be decomposed into simpler steps that can be solved via instructing the model.

---

[1] https://huggingface.co/ml6team/keyphrase-extraction-kbir-kpcrowd

| Text | Entity Extraction | Keyword Extraction | Instructing Model |
|---|---|---|---|
| John Russell Reynolds was an English physician and neurologist who made significant contributions to the field of neurology. | John Russell Reynolds, English | John Russell Reynolds, English, physician, neurologist, significant contributions, field, neurology | John Russell Reynolds, English, physician, neurologist, neurology |
| He was born in London in 1820 and studied medicine at the University of London. | London, 1820, the University of London | born, London, 1820, studied medicine, University, London | London, 1820, medicine, University of London |
| After college, he worked as a lawyer for the PGA Tour, eventually becoming the Tour's Deputy Commissioner in 1989. | the PGA Tour, Tour, 1989 | college, worked, lawyer, PGA, Tour, eventually, Tour, Deputy Commissioner | college, lawyer, PGA Tour, Deputy Commissioner, 1989 |
| He was born in Sydney in 1971 and grew up in the city's western suburbs. | Sydney, 1971 | born, Sydney, 1971, grew, city, suburbs | Sydney, 1971, western suburbs |

Table 4: Examples of concepts identified by different techniques.

| Input | Generated Sentence | Concept | Validation Question |
|---|---|---|---|
| Write an article about John Russell Reynolds | Reynolds was born in **London** in **1820** and studied **medicine** at the **University of London**. | **London** | [Y/N] Was John Russell Reynolds born in London? 
 [Wh] Where was John Russell Reynolds born? |
| | | **1820** | [Y/N] Was John Russell Reynolds born in 1820? 
 [Wh] What year was John Russell Reynolds born? |
| | | **medicine** | [Y/N] Did John Russell Reynolds study medicine? 
 [Wh] What did John Russell Reynolds study at the University of London? |
| | | **University of London** | [Y/N] Did Reynolds study medicine at the University of London? 
 [Wh] What university did John Russell Reynolds study medicine at? |

Table 5: Examples of validation questions corresponding to the identified keyphrases generated by Instructing the Model technique.

## B.3 DESIGN DECISIONS

### B.3.1 WHY THE TASK OF ADDRESSING HALLUCINATIONS IS BROKEN DOWN INTO MULTIPLE STEPS?

We note that dealing with the hallucination problem is a complex task and prior work has shown that breaking down a complex task into simpler sub-tasks helps the model in solving the task better and achieve higher performance Wei et al. (2022); Zhou et al. (2023a); Khot et al. (2023). Thus, we break down this task into individual sub-tasks which are considerably easier for the model. For the same reason, we also break down the validation procedure into several steps. We also note that creating multiple steps can increase the chances of propagation of error from one to the other; however, the individual steps in our approach are very simple, and the models perform remarkably well on these steps.

### B.3.2 WHY VALIDATION IS DONE USING THE WEB SEARCH?

Our preferred technique for retrieving knowledge is web search because the web is more likely to contain the updated knowledge in comparison to a knowledge corpus whose information can become stale, outdated, and obsolete.

### B.3.3 WHY "ACTIVE" DETECTION & MITIGATION AND NOT "POST-HOC" AFTER COMPLETE RESPONSE GENERATION?

We note that our detection and mitigation techniques can also be applied in a "posthoc" manner after complete response generation. However, it has several limitations which are addressed by our "active" approach. The "active" approach prevents the propagation of hallucinations in the subsequently generated sentences, i.e., if hallucination is detected in the initially generated sentences then it would be mitigated and course correction would be done for the subsequently generated sentences. However, the "post-hoc" approach does not provide such an opportunity of course correction. In other words, in the "active" approach, the model sees the mitigated / corrected sentences while generating the subsequent sentences; thus, its output will be more correct, coherent, and fluent. In contrast, in the "posthoc" approach, the generated sentences are based on the initially generated previous sentences and thus the mitigated sentence will not be able to influence the generation of subsequent sentences; thus, the output would not be as coherent and fluent as the active approach.

Also, applying it in a post-hoc manner will fix the sentences individually thus, redundant information could be present in multiple sentences hampering the quality of the response.

For example, for the topic "Twila Shively", the model generated "*Twila Shively is a renowned American artist and sculptor who has been creating art for over four decades. She is best known for her large-scale sculptures, which often feature abstract shapes and forms. . . .*" which is completely hallucinated.

After applying our approach in a post-hoc manner gives "*Twila Shively was an American competitive baseball player who played from 1945 through 1950 in the All-American Girls Professional Baseball League. Twila Shively is known for playing baseball. . . .*"

In contrast, active approach results in "*Twila Shively was an American competitive baseball player who played from 1945 through 1950 in the All-American Girls Professional Baseball League. She was born in Decatur, Illinois on March 20, 1922 and passed away on November 25, 1999. Twila began playing softball at the age of eight and quickly moved up in the softball ranks in Chicago. . . .*"

Thus, the active approach results in an output of much higher quality and doesn't suffer from issues such are incoherence, consistency, repetition, etc.

### B.4 WHY THE UNIT OF GENERATION IS A SENTENCE?

We select a unit as a sentence over multiple sentences and (also over just a few words instead of a sentence) because of the following reasons:

**Why not multiple sentences?** In autoregressive generation, the generation depends on the context including the model's previously generated text. Thus, if we consider multiple sentences as a unit in our approach (let's say 3 sentences) and if one of the initial sentences is hallucinated (and thus replaced with the corrected sentence), the subsequent sentences (i.e., the remaining sentences of the unit) may not stand relevant (as they were based on a sentence that has been replaced) and it may make the generation incoherent. Furthermore, the propagation of hallucination is another negative contributor as the next sentences may carry forward the hallucination of the previous incorrect sentences. Thus, the subsequent sentences in the unit would need to be regenerated. This implies that using multiple sentences as a unit may not return that benefit (that too at the extra cost of generating multiple sentences at once).

**Why not a phrase or a set of words?** We note that using a few words (i.e., a window of text) may not have sufficient information to test the correctness of the concepts in the generation. For instance, if the window is of the following words: "Rick Mahler won three gold medals and 2 silver medals at the", it doesn't have sufficient information to validate the correctness of the individual concepts.

On the other hand, a sentence typically provides richer context to validate the correctness of the concepts of the sentence.

Because of the above two reasons, we use a sentence as the unit in our method.

## B.5 Advantages of the Proposed Approach

In addition to the effectiveness and wide applicability of our approach in addressing hallucinations of LLMs (as demonstrated through extensive experiments), it has numerous other advantages:

1. It **circumvents the need for modifying the internals of LLMs** to address their hallucination problem making it a plug-and-play yet effective solution.
2. It **improves the explainability and interpretability of the LLM's output** as the generation can be attributed back to the retrieved knowledge.
3. The knowledge retrieval step **allows opportunities to use proprietary/domain-specific knowledge during the generation** process. Thus, allowing it access to the updated information.
4. Our retrieval method retrieves knowledge pertinent to the sentence and thus **enables accurate hallucination detection and mitigation**.
5. Active intervention **allows opportunities for course correction** during the generation process.

## B.6 Drawbacks of the Proposed Approach

### B.6.1 Impact on Inference Efficiency

Our approach results in improvements in the form of reduced hallucinations and thus makes the model more reliable; however, it comes at the expense of increased inference cost. However, we believe that at current time, to enable the widespread adoption of LLMs, it is more important to address their reliability and trustworthiness concerns because computational advancements are ongoing at a rapid pace. Moreover, even larger models with multi-fold times more parameters such as PaLM (540B) Chowdhery et al. (2022), Gopher (280B) Rae et al. (2021), and MT-NLG (530B) Smith et al. (2022) are also being developed which have even higher inference cost showcasing a larger focus of the community on developing better performing systems. Though it may not be a problem for all use cases, we provide a detailed discussion on it for all the steps with suggestions on their lower-cost alternatives.

**Identifying Important Concepts**: Firstly, we note the importance of this step because validating the correctness of the entire sentence at once is infeasible as a sentence can contain multiple different facets all of which can not be validated at once. In contrast, individually validating correctness corresponding to the concepts provides opportunities for accurately detecting incorrectness. Thus, if we skip this step and directly proceed to the validation step for the entire sentence then it will have limitations. For example, sentences like "*Steven Threet is best known for his time at the University of Michigan, where he was a three-year starter and led the Wolverines to a Big Ten Championship in 2008.*" contain multiple facets that need to be validated separately because a single web search may not return all the information that is required to validate the entire correctness.

This step incurs the cost of inference in which the input is the instruction (provided in Table 3) and the sentence. We mention the benefits of "instructing the model" technique in Section 2.1.1.

We discuss other lower-cost alternatives for this step below: A simple yet efficient method is to leverage a relatively smaller LLM for this step. This is feasible because identifying the concepts is an "easy" step and even smaller LLMs are typically very effective in this. Moreover, even a more smaller model such as T5 can also be finetuned for this particular task which can considerably reduce the cost. Smaller models have low inference cost (both in terms of FLOPs and latency). Furthermore, the other techniques already discussed in the paper, namely Entity Extraction and keyword extraction are other lower-cost alternatives. Specifically, the KBIR model is built on top of RoBERTa architecture which is even more efficient.

In summary, smaller models (smaller LLMs or task-specific finetuned models) can be utilized for this task to make it more efficient.

**Calculating Model's Uncertainty:** This is not a resource intensive task as it just requires calculating the score from the logit values.

**Creating Validation Question:** Similar to the first step, creating a validation query for a concept is also a task at which even smaller models (that even have only a few million parameters) do quite well. A lot of existing research on question generation uses the T5 models. Creating a validation question using an LLM requires taking the instruction (filled with the concept) (Table 3) and the sentence as input.

Another cost-effective alternative for this step is to simply mask out the selected concept from the sentence and use it as the validation query for the web search. Though, it requires some heuristics to create an appropriate validation query (such as selecting only a window of tokens on both sides of the concept after masking as the validation query, this would be required because using the entire sentence would have many different facets, and web search may not return relevant results). This would definitely make it much more efficient but it will lose effectiveness in creating "high-quality" queries pertinent to the concept and thus may not result in slight degradation in the validation procedure.

**Answering Validation Question and Mitigation Steps:** These steps are more costly than the others because they also take the retrieved knowledge as input. We note that these are crucial steps of the method. They can be made more efficient (though it will compromise the effectiveness) by combining them into a single step, i.e., validation and mitigation can be done using a single instructional prompt. However, we note that this is a relatively difficult task as compared to the previous steps and thus decomposing it into two individual steps provides better results. Thus, making this step more efficient will have tradeoffs with the performance.

Overall, these steps can be made more efficient (in terms of both computation cost and latency) using smaller LLMs or external task-specific tools. In contrast, the methodology highlighted in red in Figure 1 uses the same model for all the steps. Furthermore, we note that in resource-constrained applications, the suggested efficient alternatives can be utilized.

We present an empirical analysis of the latency where we compare the latency of all the steps of the methodology. Figure 9 shows the comparison of latency of various steps (at a sentence level). We note that the latency of the mitigation step is low as it is only conditionally called for some sentences. We show the average mitigation latency for sentences on which it is called in the Mitigation* bar. We conduct this study for 10 topics (i.e., 50 sentences) for the GPT-3.5 (text-davinci-003) model.

**Comparison of Overall Latency with the Generation**: The overall latency of the method is 2.58 times that of the regular generation (5354.20 against 2071.69).

**Why the latency of the generation step is high?** This is because for the later sentences, it also takes the context in the input.

**Why the latency of validation is high?** This is because validation procedure includes three steps (validation question creation, retrieval, and answering validation question). Furthermore, validation could be required for multiple concepts.

**What does Mitigation* represent?** Note that the mitigation step is only conditionally executed for some sentences. We show the average mitigation latency for sentences on which it is called in the Mitigation* bar.

### B.6.2  CORRECTNESS OF RETRIEVED KNOWLEDGE

Web searches can sometimes return information that is fabricated. Though we use the top web search results as our context (primarily from the reliable sources), there remains a chance that the knowledge is incorrect which can result in incorrect hallucination detection.

### B.6.3  ERROR PROPAGATION

Multiple sequential steps can increase the chances of propagation of error from one to the other; however, we note that the individual steps in our approach are very simple, and the LLMs perform remarkably well on these steps. Furthermore, our mitigation technique does not introduce new hallucinations even in the case of incorrectly detected hallucinations, i.e., false positives.

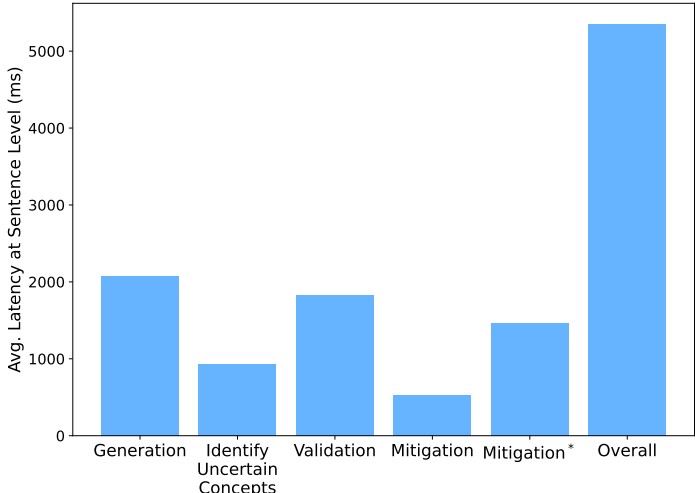

Figure 9: Comparing latency of various steps of the methodology (at a sentence level). Note that the latency of mitigation is low as it is only conditionally called for some sentences. We show the average mitigation latency for sentences on which it is called in the Mitigation* bar.

## C    EVALUATION DATA

### C.1    STATISTICS

Table 6 shows the statistics of the sentences generated by the GPT-3.5 (text-davinci-003 with **temperature 0**) model. A sentence has $\sim$ 18 words on average and each sentence has $\sim$ 3.2 key concepts that are identified by our instruction technique.

| Statistic | Mean $\pm$ Std |
|---|---|
| # Words in a Sentence | $18.6 \pm 5.55$ |
| # Key Concepts in a Sentence | $3.27 \pm 1.63$ |
| # Words in a Key Concept | $1.79 \pm 1.02$ |

Table 6: Statistics of generated sentences.

Table 7 shows examples of sentence-level and concept-level hallucination annotations.

### C.2    HUMAN ANNOTATION AND AGREEMENT WITH EXPERT ANNOTATION

We additionally compile human annotations from two annotators on randomly sampled 10 topics (50 sentences). Specifically, we asked them to mark the correctness of the sentence by searching over the web, the same annotation procedure followed for expert annotation detailed in Section 3. Cohen's kappa of the annotators with the expert annotation is 0.84 and 0.92 respectively and the kappa within themselves is 0.84. This shows the high agreement and correctness of our annotations. We note that we use our expert annotations for all the results as they are more accurate and reliable.

Since the generation is for a variety of topics of different domains and would be beyond the common knowledge of a typical human, thus, we use web search to gather the relevant information to check the correctness of the generation. Multiple web searches were required in some cases because a generation can contain multiple facets of information all of which can not be validated in a single web search.

For example, sentences like "*Steven Threet is best known for his time at the University of Michigan, where he was a three-year starter and led the Wolverines to a Big Ten Championship in 2008.*", "*Rick Mahler was a Major League Baseball pitcher who played for the Atlanta Braves, Cincinnati*

| Sentence # | Sentence | Sentence-level Correctness |
|---|---|---|
| **Sentence 1** | Eleanor Arnason is an `award-winning` `science fiction` and `fantasy` author who has been writing since the `1970s`. | Correct |
| **Sentence 2** | She is best known for her novel `A Woman of the Iron People`, which won the `James Tiptree Jr. Award` in `1991`. | Correct |
| **Sentence 3** | Her work has been praised for its exploration of `gender`, `race`, and `identity`, as well as its `imaginative world-building`. | Correct |
| **Sentence 4** | Arnason was born in `Minneapolis`, `Minnesota` in `1942`. | Hallucination |
| **Sentence 5** | She attended the `University of Minnesota`, where she earned a `degree` in `English literature`. | Hallucination |

Table 7: Examples of both sentence and concept-level annotations for the input: "Write an article about Eleanor Arnason". Annotation for correct concepts is represented in `green` while annotation for hallucinated concept is represented in `red`.

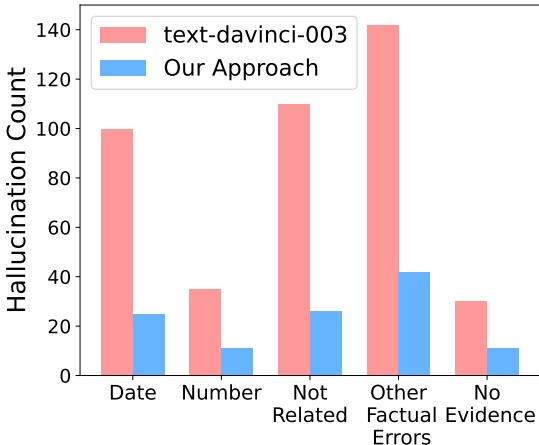

Figure 10: Comparing hallucinations across different categories.

*Reds, and St. Louis Cardinals from 1979 to 1994.*" contain multiple facets that need to be validated separately because a single web search may not return all the information that is necessary to validate the correctness of all the facets of such sentences.

## D  ACTIVE DETECTION AND MITIGATION PERFORMANCE ANALYSIS

Figure 2 compares the percentage of hallucination in the output of GPT-3.5 model and our approach. It reduces the hallucination percentage from $47.4\%$ to $14.53\%$. This proves that the active intervention during the generation process also does well in preventing the propagation of hallucination in the model's output. In Figure 10, we plot this comparison for different categories of hallucination and show that our approach does well in all the categories.

## E  RECALL OF HALLUCINATION DETECTION VS PROBABILITY THRESHOLD

Figure 11 compares the recall of hallucination detection for self-inquiry and web search techniques at different probability thresholds. **Web search considerably outperforms self-inquiry at all thresholds** and hence is better at detecting hallucinations. Selecting the probability threshold de-

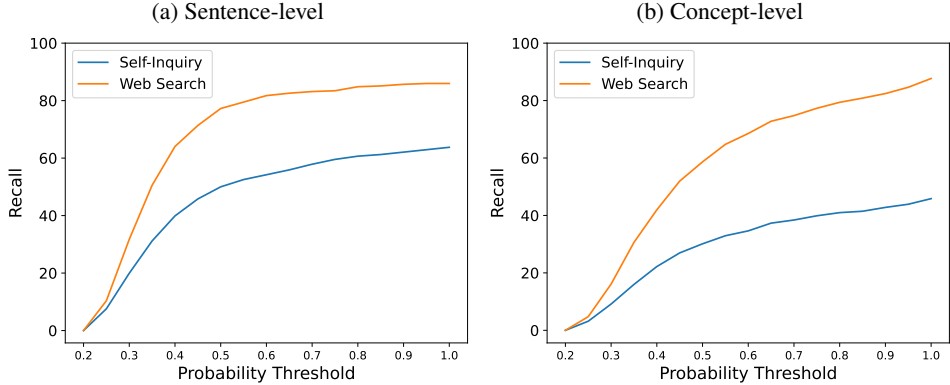

Figure 11: Recall of hallucination detection vs Probability threshold plot for Self Inquiry and web search techniques at both sentence-level and concept-level.

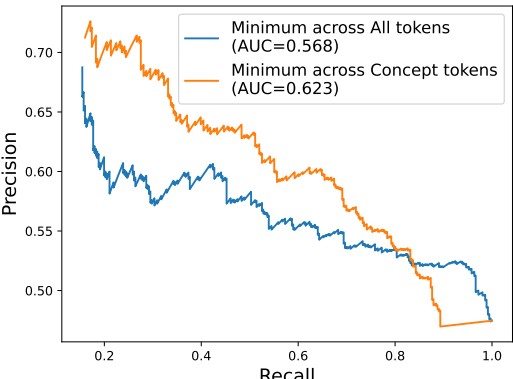

Figure 12: Demonstrating the benefit of identifying concepts from a sentence for detecting hallucinations. The figure shows precision-recall curves for the sentence level hallucination detection task corresponding to two methods that use the probabilities calculated from the logit output values. The blue curve corresponds to the technique in which we use the minimum probability across **all tokens** of the sentence and the orange curve is for the technique in which we use the minimum over **only the tokens of the identified concepts**.

pends on the tolerance level of the application. For instance, in a high-risk application domain like biomedical, we can keep a very high threshold, and in a low-risk domain like movie recommendation, we can relatively lower threshold. In this work, we use a probability threshold of 0.55. However, we note that it can be adjusted as per the application requirements.

## F  HALLUCINATION MITIGATION ANALYSIS

Tables 8 shows examples where our mitigation technique successfully mitigates the hallucinations. Table 9 shows examples where our technique fails to mitigate hallucinations. We observe that in many of the failure cases, our technique fixes some hallucinated content of the sentences but fails to fix ALL the hallucinated content from them. Furthermore, in some of the failure cases, our technique results in a sentence which is no longer hallucinated but it not completely related to the topic.

| Topic | Rick Mahler |
|---|---|
| **Original Sentence** | Mahler was born in San Diego, California in 1956. (✗) |
| **Hallucination Reason** | He was born in Austin in 1953. |
| **Modified Sentence** | Rick Mahler was born in Austin, Texas on August 5, 1953. (✔) |
| **Topic** | Matthew Aylmer, 1st Baron Aylmer |
| **Original Sentence** | Baron Aylmer, was an English naval officer and politician who served as a Member of Parliament and later as a peer in the House of Lords. (✗) |
| **Hallucination Reason** | He was in House of Commons not House of Lords. |
| **Modified Sentence** | Baron Aylmer was an Anglo-Irish Royal Navy officer and Whig politician who sat in the English and British House of Commons between 1695 and 1720. (✔) |
| **Topic** | Derek King |
| **Original Sentence** | In 2014, Derek King released his debut single "Let's Go," which quickly became a hit in Canada. (✗) |
| **Hallucination Reason** | "Let's Go" was not his song. |
| **Modified Sentence** | In 2014, Derek King released his debut mixtape Fake I.D., which featured fellow Bay Area artists Rayven Justice and IAMSU!, among others. (✔) |
| **Topic** | Marshall Manesh |
| **Original Sentence** | Marshall Manesh is an Iranian-American actor best known for his roles on the television shows How I Met Your Mother and The Middle. (✗) |
| **Hallucination Reason** | He was not in The Middle. |
| **Modified Sentence** | Marshall Manesh is an Iranian-American actor best known for his recurring roles on the television shows Will & Grace, Scrubs, Andy Barker, P.I., Hot in Cleveland, Boston Legal, and How I Met Your Mother, where he played taxi driver Ranjit. (✔) |
| **Topic** | William J. Flanagan, Jr. |
| **Original Sentence** | He is the founder and CEO of Flanagan Financial Group, a financial services firm based in New York City. (✗) |
| **Hallucination Reason** | The complete sentence is hallucinated as he is a retired US Navy admiral. |
| **Modified Sentence** | He is a retired four-star admiral who served as Commander in Chief, United States Atlantic Fleet from 1994 to 1996 and is the recipient of numerous military awards. (✔) |

Table 8: Examples of **successful** mitigation of hallucinations by our mitigation technique. Original Sentence corresponds to the sentence generated by the model and Modified Sentence corresponds to the sentence obtained on applying our technique.

| | |
|---|---|
| **Topic** | Matthew Aylmer, 1st Baron Aylmer |
| **Original Sentence** | Aylmer joined the Royal Navy in 1790 and served in the French Revolutionary Wars and the Napoleonic Wars . (✗) |
| **Hallucination Reason** | He did not serve in these wars. |
| **Modified Sentence** | Aylmer entered the Royal Navy under the protection of the Duke of Buckingham as a Lieutenant in 1678 and served in the French Revolutionary Wars and the Napoleonic Wars . (✗) |
| **Hallucination Reason** | It rectified the date but failed to correct the hallucination about wars. |
| **Topic** | K. S. Manilal |
| **Original Sentence** | Manilal was a prolific writer and translator, having written more than 50 books and translated over 100 works from English, Sanskrit , and other languages into Malayalam. (✗) |
| **Hallucination Reason** | He has not written 50 books and translated works to English and Malayalam. |
| **Modified Sentence** | Manilal was a prolific researcher and translator, having translated Hendrik van Rheede's 17th century Latin botanical treatise, Hortus Malabaricus, into English, Sanskrit , and Malayalam. (✗) |
| **Hallucination Reason** | The information about Hortus Malabaricus is correct but he translated it into English and Malayalam only and not Sanskrit. |
| **Topic** | Freddie Frith |
| **Original Sentence** | He was born in London in 1929 and began his racing career in 1951 . (✗) |
| **Hallucination Reason** | He was born in Grimsby in 1909 and began his career in 1930. |
| **Modified Sentence** | Freddie Frith was born in Grimsby, England in 1909 and began his racing career in 1935 when he won the Junior Manx Grand Prix. (✗) |
| **Hallucination Reason** | He did win the Junior Grand Prix in 1935 but he began hi career in 1930. |
| **Topic** | Harry S. Kennedy |
| **Original Sentence** | Harry S. Kennedy was an American politician who served as the 35th President of the United States from 1961 to 1963 . (✗) |
| **Hallucination Reason** | This sentence is true for John F. Kennedy not Harry S. Kennedy. |
| **Modified Sentence** | John F. Kennedy was an American politician who served as the 35th President of the United States from 1961 to 1963. (✗) |
| **Hallucination Reason** | This sentence is not hallucinated but it is not related to the topic. |

Table 9: Examples where our mitigation technique fails to mitigate complete hallucination in the generated sentence. Original Sentence corresponds to the sentence generated by the model and Modified Sentence corresponds to the sentence obtained on applying our technique.

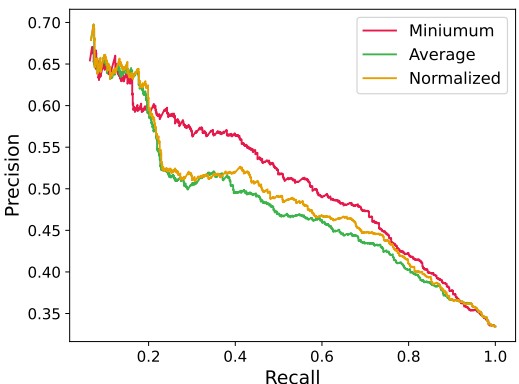

Figure 13: PR curves for the hallucination detection task (concept-level) using the three probability calculation techniques. **'Minimum' technique achieves highest AUC.**

| Question | Answer |
|---|---|
| The football manager who recruited David Beckham managed Manchester United during what timeframe? | from 1986 to 2013 |
| The Vermont Catamounts men's soccer team currently competes in a conference that was formerly known as what from 1988 to 1996? | the North Atlantic Conference |
| Ralph Hefferline was a psychology professor at a university that is located in what city? | New York City |
| What is the county seat of the county where East Lempster, New Hampshire is located? | Newport |
| Blackfin is a family of processors developed by the company that is headquartered in what city? | Norwood, Massachusetts |

Table 10: Examples of multihop questions from HotpotQA.

# G    ANALYSIS OF LOGIT OUTPUT VALUES

## G.1    BENEFIT OF IDENTIFYING CONCEPTS FROM A SENTENCE

Now, we demonstrate the benefit of identifying concepts from a sentence and leveraging the logit output values corresponding to their tokens for detecting hallucinations. To this end, we plot precision-recall curves for the hallucination detection task corresponding to two methods that use the probabilities calculated from the logit output values. The blue curve corresponds to the technique in which we use the minimum probability across **all tokens** of the sentence and the orange curve is for the technique in which we use the minimum over **only the tokens of the identified concepts**. Figure 12 shows the two curves. The orange curve achieves higher area under the precision-recall curve implying that utilizing **the probabilities of the concept tokens provides a stronger signal for hallucination** as compared to the probabilities corresponding to all the tokens.

## G.2    COMPARING PROBABILITY CALCULATION TECHNIQUES

Figure 13 shows the Precision-Recall curves for the hallucination detection task (at concept-level) using the three probability calculation techniques, i.e., Minimum, Average, and Normalized (described in 2.1.2). **The 'Minimum' technique achieves the highest area under the curve and hence is better at the hallucination detection task.**

# H  MULTI-HOP QA EXPERIMENT

## H.1  PROMPT WITH IN-CONTEXT EXAMPLES

```
Question: Which team does the player named 2015 Diamond Head Classic's MVP
play for?

Answer: The 2015 Diamond Head Classic's MVP was Buddy Hield. Buddy Hield
plays for the Sacramento Kings of the NBA. So, the answer is Sacramento
Kings.

Question: The Dutch-Belgian television series that "House of Anubis" was
based on first aired in what year?

Answer: "House of Anubis" was based on is Het Huis Anubis. Het Huis Anubis
was first aired in 2006. So, the answer is 2006.

Question: What is the birth date of the coach of the 2014-15 Michigan
Wolverines men's basketball team?

Answer: The coach of the 2014-15 Michigan Wolverines men's basketball team
was John Beilein. John Beilein's birth date is February 5, 1953.

Question: {question}
Answer:
```

Prompt with in-context examples used for prompting the model to answer 'bridge' multi-hop questions step by step.

## H.2  EXAMPLES OF MULTI-HOP BRIDGE QUESTIONS

Table 10 shows examples of multi-hop bridge questions from HotpotQA dataset.

## H.3  EXAMPLES OF MODEL RESPONSES

Table 11 shows examples of responses generated using our approach for multi-hop bridge questions.

# I  FALSE PREMISE QA EXPERIMENT

Table 12 shows examples of false premise and true premise question pairs. Table 15 shows responses generated on a few false premise questions by the GPT-3.5 (text-davinci-003) model, GPT-3.5 (text-davinci-003) using the retrieved knowledge as context, and our approach.

**Efficacy of Question Rectification:** We analyze the performance of our approach in rectifying questions; **it successfully repairs 76% false premise questions while not incorrectly modifying any true premise question.** Though this step makes modifications in a small number of true premise questions (6 instances), it does not change their semantics as shown in Table 14. Not incorrectly modifying a true premise question is an important characteristic of this approach.

**Question**: *Kent Dairy Round Barn is near the city in what Montana county?*
**Response**: The Kent Dairy Round Barn is located near Red Lodge, Montana. Red Lodge is located in Carbon County, Montana. So, the answer is `Carbon County` .

**Question**: *What political party was the 6th governor of Hawaii that passed bills to help with global warming in Hawaii?*
**Response**: The 6th governor of Hawaii was Linda Lingle. Linda Lingle was a member of the Republican Party and passed bills to help with global warming in Hawaii. So, the answer is `Republican Party` .

**Question**: *In what city is the mortgage lending company, founded by Dan Gilbert, headquartered?*
**Response**: The mortgage lending company founded by Dan Gilbert is Quicken Loans. Quicken Loans is headquartered in Detroit, Michigan. So, the answer is `Detroit, Michigan` .

**Question**: *Actor David Lee Stenstrom played the character Waldo the inventor in a TV show that ran on Nickelodeon during what yeras?*
**Response**: The TV show was Out of Control (1984). David Lee Stenstrom played the character Waldo the inventor in the show. The show ran on Nickelodeon from 1984 to 1985. So, the answer is `1984 to 1985` .

**Question**: *What music school did the singer of The Search for Everything: Wave One attend?*
**Response**: The singer of The Search for Everything: Wave One is John Mayer. John Mayer attended the Berklee College of Music in Boston, Massachusetts. So, the answer is `Berklee College of Music` .

Table 11: Examples of responses generated using our approach for multihop bridge questions.

| False Premise Question | True Premise Question |
|---|---|
| Why does Mars have **three** moons? | Why does Mars have two moons? |
| Why are golf balls **bigger** than basketballs? | Why are golf balls smaller than basketballs? |
| What are some papers on the relationship between homeschooling and **neuroplasticity**? | What are some papers on the relationship between homeschooling and college preparedness? |
| Why do migrant workers **never leave** their home? | Why do migrant workers leave their home? |
| When a diver swims deeper, why does the water **pressure declines**? | When a diver swims deeper, why does the water pressure increases? |
| Why does Mars have **higher gravity** than Earth? | Why does Mars have lower gravity than Earth? |
| Why is **baseball the most popular sport** in the world? | Why is soccer the most popular sport in the world? |
| How does **Earth make up over 99% of the solar system's mass**? | How does Sun make up over 99% of the solar system's mass? |
| Why does **Helium have atomic number of 1**? | Why does Hydrogen have atomic number of 1? |

Table 12: Examples of 'false premise' questions and their corresponding 'true premise' counterparts.

| Step | Prompt |
|---|---|
| I | `Context: {search results} Question: {question} Based on the above context, does the above question make factually correct assumptions?` |
| II | `Context: {response_of_Step_I} Question: {question} Rectify the incorrect information in the given question based on the context.` |

Table 13: Instructional Prompts for rectifying the false premise questions.

| Original Question | After Modification |
|---|---|
| **False Premise Questions** | |
| Why does Mars have three moons? | Why does Mars have two moons? (✓) |
| Why are golf balls bigger than basketballs? | Why are golf balls smaller than basketballs? (✓) |
| What are some papers on the relationship between homeschooling and neuroplasticity? | What are some papers on the relationship between homeschooling and learning outcomes? (✓) |
| Why USA has the lowest happiness index? | What factors have contributed to the decline in happiness among Americans? (✓) |
| How many metres does a typical apple weigh? | How many grams or ounces does a typical apple weigh? (✓) |
| Why do gases have a particular shape? | Why do gases not have a definite volume or shape? (✓) |
| Why do migrant workers never leave their home? | Why do migrant workers leave their home? (✓) |
| When a diver swims deeper, why does the water pressure declines? | When a diver swims deeper, why does the water pressure increase? (✓) |
| Why does Mars have higher gravity than Earth? | Why does Mars have weaker gravity than Earth? (✓) |
| Why do sprinters run at slower speeds than marathon runners? | Why do sprinters sometimes run at faster speeds than marathon runners? (✓) |
| Why do all rabbits have red eyes? | Why do some rabbits have red eyes? (✓) |
| Why does Helium have atomic number of 1? | Why does Helium have atomic number of 2? (✓) |
| Why does Bangladesh have the highest population in the world? | Why does Bangladesh have the highest population growth rate in the world? (✗) |
| Why are tigers' eggs bigger than chicken's eggs? | Why do some breeds of chickens lay larger eggs than others? (✗) |
| **True Premise Questions** | |
| Why gases are shapeless? | Why are gases shapeless? (✓) |
| How did USA become a developed country? | How did the United States become a developed country (✓) |
| Why Afghanistan has a low happiness index? | What factors contribute to Afghanistan's low happiness index? (✓) |
| Why are golf balls smaller than basketballs? | Why are golf balls typically smaller than basketballs? (✓) |
| How were the 2020 USA presidential election? | What were the results of the 2020 USA presidential election? (✓) |

Table 14: Examples of original questions (both false premise and true premise) and the questions after rectification. ✗and ✓indicate that the modified question is incorrect and correct, respectively.

| | |
|---|---|
| **False Premise Question** | *Why are golf balls **bigger** than basketballs?* |
| **GPT-3.5** | Golf balls are bigger than basketballs because they need to be able to travel farther distances and have a larger surface area to create more spin and control. The larger size also helps to reduce air resistance, allowing the ball to travel farther. (✗) |
| **GPT-3.5 w/ Knowledge** | Golf balls are bigger than basketballs because they need to be heavier and denser in order to travel farther when hit with the same amount of force. The larger size allows for more mass-density, which gives the ball more momentum and distance when hit. (✗) |
| **Our Approach** | Golf balls are smaller than basketballs because they are optimized for golfers to provide the most ball velocity and distance possible, along with a controllable spin rate. The diameter of a golf ball must reach 1.68 inches, or 42.67 mm, in diameter, and a weight of no more than 1.62 ounces, or 45. (✓) |
| **False Premise Question** | *Why do deserts have **higher** soil moisture than grasslands?* |
| **GPT-3.5** | Deserts have higher soil moisture than grasslands because they receive more rainfall. Deserts typically receive less rainfall than grasslands, but the rain that does fall is more concentrated and intense. This causes the soil to absorb more moisture than in grasslands, where the rainfall is more spread out and less intense. (✗) |
| **GPT-3.5 w/ Knowledge** | Deserts have higher soil moisture than grasslands because they receive more precipitation than grasslands. Deserts are located in areas where the demand for water by the atmosphere (evaporation) and plants (transpiration) is much greater than precipitation. This means that the soil in deserts is able to retain more moisture than the soil in grasslands. (✗) |
| **Our Approach** | Deserts have lower soil moisture than grasslands because they have less water vapor in the air, which makes it harder to trap heat or cold in a desert. Additionally, deserts typically have less precipitation than grasslands, which further contributes to lower soil moisture levels. (✓) |
| **False Premise Question** | *Why is it more **difficult** to swim in seawater than fresh river water?* |
| **GPT-3.5** | Seawater is more dense than fresh river water due to its higher salt content. This increased density makes it more difficult to swim in because it requires more effort to move through the water. Additionally, the salt content of seawater can cause skin irritation and dehydration, making it more uncomfortable to swim in. (✗) |
| **GPT-3 w/ Knowledge** | It is more difficult to swim in seawater than fresh river water because seawater is more dense than fresh river water. This means that the buoyancy of the swimmer is reduced in seawater, making it more difficult to stay afloat. Additionally, the salt content in seawater can cause skin irritation and dehydration, making it more difficult to swim. (✗) |
| **Our Approach** | It is easier to swim in seawater than fresh river water because seawater is denser than fresh water due to its higher mineral content, including sodium, chloride, sulphate, magnesium and calcium. This increased density provides more buoyancy, making it easier for swimmers to stay afloat. (✓) |

Table 15: Comparing responses generated on a few false premise questions by the GPT-3.5 model, GPT-3.5 moel leveraging the retrieved knowledge as context, and our approach.

## J EFFECTIVENESS OF THE METHOD BEYOND THE FIRST FIVE GENERATED SENTENCES

Our study on the article generation task is conducted on the first five generated sentences. After applying our method, the correctness at sentence number level (averaged over all the inputs) is as follows (Sentence 1: 90.0%, Sentence 2: 82.67%, Sentence 3: 86.67%, Sentence 4: 82.67%, Sentence 5: 85.34%). These values are indeed close and do not considerably reduce as the sentence number increases. With this result, we show that our method of active detection and mitigation successfully mitigates the hallucination throughout the generation (not restricted to any specific sentence number). Furthermore, it shows that the ability to address hallucinations does not considerably diminish as the sentence number increases. Thus, even increasing the number of sentences is not expected to considerably impact the improvement that our method would bring

## K OTHER APPLICATIONS OF OUR APPROACH

Our approach has utility in a variety of other applications also such as Abstractive Summarization and Claim Verification. In abstractive summarization where the generated summary has been shown to be often hallucinated Cao et al. (2022); Zhao et al. (2020); Chen et al. (2021) can be improved using our approach. Here, the relevant knowledge during validation will be retrieved from the original document instead of the web. Our approach can be adapted for the claim verification task also as we can first identify the key sub-claims and then verify each sub-claim using the validation procedure. Here, the mitigation step will also be useful for providing explanations behind the model's decision. We leave exploring these other usecases of our approach for future work.

