# OpenReview forum: "A Stitch in Time Saves Nine: Detecting and Mitigating Hallucinations of LLMs by Actively Validating Low-Confidence Generation"
_ICLR.cc/2024/Conference — ICLR 2024 Conference Withdrawn Submission_

### Official Review · Reviewer_XFUz · 2023-10-28

**Soundness:** 2 fair
**Presentation:** 3 good
**Contribution:** 2 fair
**Rating:** 5
**Confidence:** 4

**Summary:**

The paper proposes a multi-step procedure to reduce hallucinations in the output of LLMs.
The procedure first identifies key concepts in a sentence, filters them to uncertain ones using the model output logits, retrieves information using web search, validates the concepts by prompting, and finally corrects the output sentence by prompting with the retrieved context as support. The main experiments are conducted on article generation using a closed dataset (promised to be released after publication) using GPT-3.5 and Vicuna 1.5.

**Strengths:**

- The multi-step approach presented is generally sound. The approach can use black-box models hidden behind an API, and several possible solutions for each individual step are presented and evaluated to some extent.
- The experiments are primarily based on GPT-3.5, but there are also experiments with Vicuna-1.5 to validate the results. In addition, the use of an open model supports easier reproduction and improves the overall accessibility of the presented methodology.
- Hallucinations are a relevant problem with current LLMs and are a limitation to their general applicability.

**Weaknesses:**

- The proposed multi-step approach is likely to increase generation latency significantly. While this is noted superficially, and an improvement for one of the many steps is roughly sketched out, an in-depth discussion is missing - in particular, there are no experiments or theoretical discussions about the overall latency. I am not of the opinion that high latency is a problem for all use cases, but it would be important to have a proper discussion about this limitation and where it is a problem.
- The overall experimental design is not described in sufficient detail. In particular, it is not clear how the data used for Section 3.1 relate to those used for Sections 3.2 and 3.3. If they were the same data, I would be concerned about the reliability of the results in the later sections, since the hyperparameters of each step, such as the aggregation method used to obtain concept uncertainty, are chosen to maximize the metrics in a data set.
- It is not clear to what extent retrieval alone explains the reduction in hallucinations. Given that the proposed method uses (multiple) web search queries, a natural baseline would be to consider the article generation task based on retrieved facts about the article topic, which would have some favorable properties (e.g., lower latency, less technical complexity) compared to the proposed multi-step approach. A proper ablation/evaluation against this baseline could help to delineate this effect.
- Some of the design decisions seem to be taken quite ad-hoc; for instance, the choice of a method for key concept identification seems to be based on qualitatively looking at a few examples (Table 4, Section B.1)

**Questions:**

- Table 1: How can sentence-level recall (85.96) be smaller than concept-level recall (87.68) if a sentence is considered hallucinated as soon as a single concept is hallucinated?
- Section 3 describes the data selection process. In particular, the topics for article generation are selected based on the longest articles in WikiBio or Wikipedia. I would expect this selection strategy to select topics with high prevalence in most LLMs training data: either because they train directly on Wikipedia, but also because long Wikipedia articles are likely to be about a topic of general interest with high coverage in web data as well. How does this affect hallucination, or in other words, how representative are the results for hallucination detection and mitigation based on these topics?
- Regarding labeling hallucinations, how do you handle sentences that are correct given the content of previously hallucinated sentences?
- Your methodology works on the unit of a sentence. The (initial) output of the model would be a paragraph/full text. How do you segment it into sentences? What is your motivation for the sentence unit? Since you are processing the key concepts sequentially, do you need the sentence separation? Or could the approach work directly on the whole paragraph?
- Since the uncertainty calculation serves as a filter on the concepts sent for verification, I wonder about the relative importance of precision and recall. Intuitively, I would expect this to be a recall-oriented scenario, but the decision seems to be based on the area under the ROC.
- Section 2.1.5.
  > However, running this in parallel would require starting multiple threads which may not be supported by all machines.

  shouldn't this be easily solved by batching the requests?
- Section 3.1.1.: Choosing more descriptive names than `A`, `B`, `C`, and `D`, such as `YY`, `YN`, `NY`,  and`NN`, or directly using the conditional probability notation $p(H | H)$, $p(H | \neg H)$, ... would greatly improve the readability of the plots and discussion.
- Figure 5: A bar chart would be more appropriate here; I would also be interested in the confidence/dependence on the selected sentences. One way to study this would be to use bootstrapping.
- In general, I disliked the overuse of bold type, as it reduced readability quite a bit. The same goes for the use of free-floating figures. An extreme example is page 6.
- Table 1: Accuracy seems to be in $[0, 1]$, while precision and recall are given in $[0, 100]$ (i.e., in percent).
- Section 3.2, "Mitigation" + Table 2: The numbers in the text do not seem to match those in the table.
- For Section 4.2 / QA, it is not clear how the multi-step approach works at all? I would guess that the answer is always a single sentence (or even a sentence fragment), so how is the iterative sentence-level method applied?
- Section 4.3
  > Importantly, we also show that our approach does not incorrectly modify a true premise question.

  where is this shown?

- Appendix A: Related Work; It would be nice if the ones listed under "concurrent work" were part of the main paper description, as they seem to be the most related.
- B.3.2:
 > Our preferred technique for retrieving knowledge is web search because the web is more likely to
contain the updated knowledge in comparison to a knowledge corpus whose information can become
stale, outdated, and obsolete.

  If (one of) the main reasons for hallucinations is outdated knowledge, wouldn't we notice that the model uncertainty does not reflect this properly, i.e. the model is very certain about its outdated knowledge?

- G.1: The sentence-level baseline uses the minimum probability over all tokens; I think it would make more sense to consider the other aggregations as well.

#### Minor Remarks
- Section 2, first paragraph: there is an inconsistent use of "Section" vs. "section"
- Section 2, first paragraph, line 3: typo: "shwon" should be "shown"
- Section 2.1.2; "normalized product of probabilities" seems to be equivalent to the geometric mean of probabilities; the latter may be the preferred term for some, so it would be nice to make this more clear (e.g., footnote, rename, ...)

---

> ### Author Response · Authors · 2023-11-15
>
> We thank and sincerely appreciate the detailed comments by the reviewer.
>
> **1) A Detailed Discussion on Latency:**
>
> We note that our approach results in improvements in the form of reduced hallucinations and thus makes the model more reliable (in addition to all the other advantages specified in B.4); however, it comes at the expense of increased inference cost.
> We completely agree with the reviewer’s point that though high latency may not be a problem for all use cases, it would be important to have a detailed discussion on it. We have briefly discussed this point in Appendix B.5.1. Here, we provide a detailed discussion on it for all the steps with suggestions on their lower-cost alternatives. We will also include it in the revision.
>
> **Identifying Important Concepts:**
> Firstly, we note the importance of this step because validating the correctness of the entire sentence at once is infeasible as a sentence can contain multiple different facets all of which can not be validated at once. In contrast, individually validating correctness corresponding to the concepts provides opportunities for accurately detecting incorrectness.
> Thus, if we skip this step and directly proceed to the validation step for the entire sentence then it will have limitations. For example, sentences like “​Steven Threet i​s best known for his time at the University of Michigan, where he was a three-year starter and led the Wolverines to a Big Ten Championship in 2008.” contain multiple facets that need to be validated separately because a single web search may not return all the information that is required to validate the entire correctness.
>
> This step incurs the cost of inference in which the input is the instruction (provided in Table 3) and the sentence. We mention the benefits of “instructing the model” technique in Section 2.1.1.
>
> We discuss other lower-cost alternatives for this step below:
> A simple yet efficient method is to leverage a relatively smaller LLM for this step. This is feasible because identifying the concepts is an “easy” step and even smaller LLMs are typically very effective in this. Moreover, even a more smaller model such as T5 can also be finetuned for this particular task which can considerably reduce the cost. Smaller models have low inference cost (both in terms of FLOPs and latency).
> Furthermore, the other techniques already discussed in the paper, namely Entity Extraction and keyword extraction are other lower-cost alternatives. Specifically, the KBIR model is built on top of RoBERTa architecture which is even more efficient.
> In summary, smaller models (smaller LLMs or task-specific finetuned models) can be utilized for this task to make it more efficient.
>
> **Calculating Model’s Uncertainty**: This is not a resource intensive task as it just requires calculating the score from the logit values.
>
> **Creating Validation Question**: Similar to the first step, creating a validation query for a concept is also a task at which even smaller models (that even have only a few million parameters) do quite well. A lot of existing research on question generation uses the T5 models.
> Creating a validation question using an LLM requires taking the instruction (filled with the concept) (Table 3) and the sentence as input.
>
> Another cost-effective alternative for this step is to simply mask out the selected concept from the sentence and use it as the validation query for the web search. Though, it requires some heuristics to create an appropriate validation query (such as selecting only a window of tokens on both sides of the concept after masking as the validation query, this would be required because using the entire sentence would have many different facets, and web search may not return relevant results). This would definitely make it much more efficient but it will lose effectiveness in creating “high-quality” queries pertinent to the concept and thus may not result in slight degradation in the validation procedure.
>
> **Answering Validation Question & Mitigation**: These steps are more costly than the others because they also take the retrieved knowledge as input. We note that these are crucial steps of the method.
> They can be made more efficient (though it will compromise the effectiveness) by combining them into a single step, i.e., validation and mitigation can be done using a single instructional prompt. However, we note that this is a relatively difficult task as compared to the previous steps and thus decomposing it into two individual steps provides better results.
> Thus, making this step more efficient will have tradeoffs with the performance.
>
> Overall, these steps can be made more efficient (in terms of both computation cost and latency) using smaller LLMs or external task-specific tools. In contrast, the methodology highlighted in red in Figure 1 uses the same model for all the steps. Furthermore, we note that in resource-constrained applications, the suggested efficient alternatives can be utilized.

---

> > ### Author Response · Authors · 2023-11-15
> >
> > **2) Clarification on application of our method in Multihop QA**: Note that multi-hop questions such as “who is the wife of the president of USA?” typically require multiple steps to reach the answer. However, a system can directly produce the answer also (this corresponds to the first baseline in FIgure 7). Motivated by the success of CoT and other prominent prompting techniques, another strategy is to enable the model to answer the question in a step-by-step manner, in our running example, it would be “The president of USA is Joe Biden. Joe Biden’s wife is Jill Biden, so the answer is Jill Biden”. Such decomposition and step by step answering techniques have been shown to be very effective. Because it is autoregressive generation, a mistake in a step can propagate in future steps and result in incorrect final answer.
> >
> > We improve on this with our active detection and mitigation technique. Here, we consider one step as a unit (same as sentence in the generation task).
> > To this end, we prompt the model and provide in-context examples demonstrating it to answer a given multi-hop question step by step. Appendix H shows the corresponding prompt used for this purpose. Specifically, for a test question, the model generates the answer in multiple steps (one step at a time) and for each step, we apply our technique in which we detect and mitigate the hallucination, and then proceed to generate the next step.
> >
> > We compare the performance of various baselines in Figure 7 and show the effectiveness of our method.
> >
> > --------------
> >
> > **3) Difference Between Sentence-level Recall and Concept-level Recall:** Please note that the concept-level recall is calculated based on concept-level annotations (and not sentence-level annotations). A sentence typically has multiple concepts (3.27 on average) and any of those concepts could be hallucinated or not-hallucinated. Thus, the sentence-level annotations are different from concept-level annotations. For example, if a hallucinated sentence has 3 concepts, it could be hallucinating on one or more concepts. Thus, for sentence-level, there is one (sentence, annotation) pair; however, for concept-level there would be three (concept, annotation) pairs. This justifies the difference in recall values in Table 1.
> >
> > --------------
> >
> > **4) Clarification on Percentage Numbers specified in Section 3.2:** Table 2 shows the percentage of the four scenarios for (Before Modification, After Moficiation)
> >
> > (Halucinated, Not Halucinated) - 40.81%
> >
> > (Halucinated, Halucinated) - 30.04%
> >
> > (Not Halucinated, Not Halucinated) - 28.26%, and
> >
> > (Not Halucinated, Halucinated) - 0.89%).
> >
> > In Section 3.2, we mention that “It successfully mitigates the hallucination on 57.6% of the correctly detected hallucinations (True Positives)”. Therefore, this number corresponds to (40.81/ (40.81 + 30.04) = 57.6%).
> >
> > --------------
> >
> > **5) Justification of selecting the instruction technique for the concept identification step:**
> > In addition to the qualitative comparison, we note that the instruction technique has several advantages over our other alternative techniques proposed in Section 2.1.1. Firstly, it does not require an external task-specific tool (like the entity extraction model or the keyword extraction model). Secondly, this is an “easy” task for the LLMs which can be observed from even the smaller LLMs that do quite well on this task. Finally, we show its effectiveness in the overall task of reducing hallucinations. Also, we would like to note that we have proposed multiple alternatives for all the steps of our approach for generality and completeness. Scenarios in which a more efficient solution is required, the other alternatives like a smaller LLM or specially finetuned model can be used as a substitute (as discussed (in Response 1) above).
> >
> > -------------
> >
> > **6) Examples of Modifications of the false premise questions:** Table 14 in Appendix shows examples of original questions (both false premise and true premise) and the questions after modification.
> >
> > --------
> >
> > **7) Regarding labeling hallucinations, how do you handle sentences that are correct given the content of previously hallucinated sentences?**
> > A sentence that is correct (i.e., does not have any hallucinations) is marked as correct. A correct sentence followed by a hallucinated sentence is also a possibility (out of the total four possibilities considered in Figure 4).

---

> > > ### Author Response · Authors · 2023-11-15
> > >
> > > **8) Motivation for using a sentence as a unit:**
> > > We select a unit as a sentence over multiple sentences and (also over just a few words instead of a sentence) because:
> > >
> > > Why not multiple sentences:
> > > In autoregressive generation, the generation depends on the context including the model’s previously generated text. Thus, if we consider multiple sentences as a unit in our approach (let’s say 3 sentences) and if one of the initial sentences is hallucinated (and thus replaced with the corrected sentence), the subsequent sentences (ie. the remaining sentences of the unit) may not stand relevant (as they were based on a sentence that has been replaced) and it may make the generation incoherent. Furthermore, the propagation of hallucination is another negative contributor as the next sentences may carry forward the hallucination of the previous incorrect sentences. Thus, the subsequent sentences in the unit would need to be regenerated. This implies that using multiple sentences as a unit may not return that benefit (that too at the extra cost of generating multiple sentences at once).
> > >
> > > Why not use a few words as a unit:
> > > We note that using a few words (ie. a window of text) may not have sufficient information to test the correctness of the concepts in the generation. For instance, if the window is of the following words: “Rick Mahler won three gold medals and 2 silver medals at the”, it doesn’t have sufficient information to validate correctness of individual concepts. On the other hand, a sentence typically provides richer context to validate the correctness of the concepts of the sentence.
> > >
> > > Because of the above reasons, we use a sentence as the unit
> > >
> > > ---
> > >
> > > **9) Data Used for Performance Evaluation vs Data for hyperparameters**:
> > > The data used for performance evaluation of the approach is different from the data used for hyperparameters. The hyperparameter in the approach - the probability score threshold for the concepts is calculated using the plot created for the concepts of the raw generation. On the other hand, our approach generates output using our active detection and mitigation methodology. Thus, the performance evaluation of the approach does not have any kind of leakage as the hyperparameter selection is based on a different output. This is further demonstrated by its effectiveness in other experimental settings, namely, MultihopQA and FalsePremiseQA
> > >
> > > ---
> > >
> > > **10) Effectiveness of Retrieval alone**
> > >
> > > We agree with the reviewer that retrieval alone has advantages like lower latency and less technical complexity than the proposed approach. However, as we show in our work our approach results in considerably higher performance. For a fair comparison, we have compared the performance of retrieval alone with our active intervention approach. We will also underline the advantages of our active intervention method over the retrieval alone method.
> > >
> > > Figure 7 shows this comparison for the MultihopQA settings. Specifically, using the retrieved knowledge alone (retrieved using the question as the search query), the model’s hallucination is at 38%. Using our approach of active intervention the hallucination is at 26%. We attribute our performance improvement to the active correction in the intermediate steps which eventually leads to improved answers.
> > >
> > > Similarly, in the false premise QA setting, we show this comparison in Figure 8. We note that in this case, the improvement is even larger (76% vs 24%). This is because of a recently studied concept of sycophancy, where LLMs tend to generate responses that favor the user’s perspective rather than providing correct or truthful answers, which can result in hallucinations. Our approach addresses this problem and reduces the hallucination.
> > >
> > > Advantages of our active intervention over the retrieval alone baseline:
> > >
> > > Firstly, active retrieval retrieves the knowledge that is pertinent to the current sentence in the generation. In contrast, single retrieval retrieves only once and does not have the opportunity of retrieving knowledge pertinent to the current sentence.
> > >
> > > Also, active intervention allows opportunities for course correction during the generation process i.e. if a sentence is hallucinated then it is fixed and then the subsequent sentences are generated. This prevents the propagation of hallucinations and also drives the generation in the right direction.
> > >
> > > Furthermore, single retrieval can constrain the generation to be dependent on what has been retrieved initially. In contrast, active intervention allows the model to follow its course of generation and retrieve the knowledge based on that, unlike single retrieval where the generation is based on the retrieved knowledge
> > >
> > > ---
> > >
> > > **Typos and Presentation-related suggestions**: Thanks for all the valuable suggestions. We will definitely incorporate them in the revision.

---

> > > > ### Comment · Reviewer_XFUz · 2023-11-16
> > > >
> > > > Thank you for your reply. Due to the excessive length and changed order, it was a bit difficult to match your responses to my points. I have tried to include what I thought was relevant, but feel free to correct me where I have misaligned them. Some of my points have been clarified, but others are still open questions for me.
> > > >
> > > > >>The proposed multi-step approach is likely to increase generation latency significantly. While this is noted superficially, and an improvement for one of the many steps is roughly sketched out, an in-depth discussion is missing - in particular, there are no experiments or theoretical discussions about the overall latency. I am not of the opinion that high latency is a problem for all use cases, but it would be important to have a proper discussion about this limitation and where it is a problem.
> > > > >
> > > > >**1) A Detailed Discussion on Latency:** [...]
> > > >
> > > > Thank you for providing a discussion in your comment, and for pointing out lower-cost alternatives. I still think that this discussion should be an important part of the paper, and should be supported by experimental evidence of the impact on latency, as this provides a basis for deciding the trade-off between better answers and lower latency. In particular, I would also like to see how it compares to alternative approaches such as those mentioned by reviewers `pdV3` and `EApd`.
> > > >
> > > >
> > > > >> For Section 4.2 / QA, it is not clear how the multi-step approach works at all? I would guess that the answer is always a single sentence (or even a sentence fragment), so how is the iterative sentence-level method applied?
> > > > >
> > > > >**2) [...]
> > > >
> > > > Thank you for your clarification.
> > > >
> > > > >> Table 1: How can sentence-level recall (85.96) be smaller than concept-level recall (87.68) if a sentence is considered hallucinated as soon as a single concept is hallucinated?
> > > > >
> > > > >**3) [...]
> > > >
> > > > Thank you for your clarification.
> > > >
> > > > >>Section 3.2, "Mitigation" + Table 2: The numbers in the text do not seem to match those in the table.
> > > > >
> > > > >**4) [...]
> > > >
> > > > Thank you for your clarification.
> > > >
> > > > >>Some of the design decisions seem to be taken quite ad-hoc; for instance, the choice of a method for key concept identification seems to be based on qualitatively looking at a few examples (Table 4, Section B.1)
> > > > >
> > > > >**5) Justification of selecting the instruction technique for the concept identification step:** In addition to the qualitative comparison, we note that the instruction technique has several advantages over our other alternative techniques proposed in Section 2.1.1. Firstly, it does not require an external task-specific tool (like the entity extraction model or the keyword extraction model). Secondly, this is an “easy” task for the LLMs which can be observed from even the smaller LLMs that do quite well on this task. Finally, we show its effectiveness in the overall task of reducing hallucinations. Also, we would like to note that we have proposed multiple alternatives for all the steps of our approach for generality and completeness. Scenarios in which a more efficient solution is required, the other alternatives like a smaller LLM or specially finetuned model can be used as a substitute (as discussed (in Response 1) above).
> > > >
> > > > I do agree that an instruction-tuned LLM isprobably capable of identifying key concepts and thus an appropriate choice, as are the other two approaches suggested. My point is that I would have preferred to see experimental support for your choices and observations, e.g.,
> > > >
> > > > >It shows that the entity extraction model often misses many important concepts while the keyword extraction model identifies a number of insignificant concepts also
> > > >
> > > > rather than referring to a table with a few qualitative examples.
> > > >
> > > > > > Section 4.3
> > > > > > > Importantly, we also show that our approach does not incorrectly modify a true premise question.
> > > > > >  where is this shown?
> > > > >
> > > > > **6) Examples of Modifications of the false premise questions:** Table 14 in Appendix shows examples of original questions (both false premise and true premise) and the questions after modification.
> > > >
> > > > This statement is solely based on the five examples shown in Table 14?

---

> > > > > ### Comment · Reviewer_XFUz · 2023-11-16
> > > > >
> > > > > > > Regarding labeling hallucinations, how do you handle sentences that are correct given the content of previously hallucinated sentences?
> > > > > >
> > > > > > **7) Regarding labeling hallucinations, how do you handle sentences that are correct given the content of previously hallucinated sentences?** A sentence that is correct (i.e., does not have any hallucinations) is marked as correct. A correct sentence followed by a hallucinated sentence is also a possibility (out of the total four possibilities considered in Figure 4).
> > > > >
> > > > > What I meant was something like
> > > > > >An apple weighs 5,000 kg. This is more than a typical horse.
> > > > >
> > > > > Would the second sentence be considered correct, or incorrect?
> > > > >
> > > > > > > Your methodology works on the unit of a sentence. The (initial) output of the model would be a paragraph/full text. How do you segment it into sentences? What is your motivation for the sentence unit? Since you are processing the key concepts sequentially, do you need the sentence separation? Or could the approach work directly on the whole paragraph?
> > > > > >
> > > > > > **8) Motivation for using a sentence as a unit:** [...]
> > > > >
> > > > > Thank you for your clarification.
> > > > >
> > > > > > > The overall experimental design is not described in sufficient detail. In particular, it is not clear how the data used for Section 3.1 relate to those used for Sections 3.2 and 3.3. If they were the same data, I would be concerned about the reliability of the results in the later sections, since the hyperparameters of each step, such as the aggregation method used to obtain concept uncertainty, are chosen to maximize the metrics in a data set.
> > > > > >
> > > > > > **9) Data Used for Performance Evaluation vs Data for hyperparameters**: The data used for performance evaluation of the approach is different from the data used for hyperparameters. The hyperparameter in the approach - the probability score threshold for the concepts is calculated using the plot created for the concepts of the raw generation. On the other hand, our approach generates output using our active detection and mitigation methodology. Thus, the performance evaluation of the approach does not have any kind of leakage as the hyperparameter selection is based on a different output. This is further demonstrated by its effectiveness in other experimental settings, namely, MultihopQA and FalsePremiseQA
> > > > >
> > > > > With
> > > > >
> > > > > > The hyperparameter in the approach - the probability score threshold for the concepts is calculated using the plot created for the concepts of the raw generation.
> > > > >
> > > > > do you mean that you use, for example, the prompt shown in Figure 1, "Write an article on Rick Mahler," to get a generation of the LLM, and then look at a plot of probabilities for each token? How do you decide which are the key concepts? Do you label them manually? Or do you use one of the other two approaches?

---

> > > > > > ### Comment · Reviewer_XFUz · 2023-11-16
> > > > > >
> > > > > > > > It is not clear to what extent retrieval alone explains the reduction in hallucinations. Given that the proposed method uses (multiple) web search queries, a natural baseline would be to consider the article generation task based on retrieved facts about the article topic, which would have some favorable properties (e.g., lower latency, less technical complexity) compared to the proposed multi-step approach. A proper ablation/evaluation against this baseline could help to delineate this effect.
> > > > > > >
> > > > > > > **10) Effectiveness of Retrieval alone**
> > > > > > > We agree with the reviewer that retrieval alone has advantages like lower latency and less technical complexity than the proposed approach. However, as we show in our work our approach results in considerably higher performance. For a fair comparison, we have compared the performance of retrieval alone with our active intervention approach. We will also underline the advantages of our active intervention method over the retrieval alone method.
> > > > > > > Figure 7 shows this comparison for the MultihopQA settings. Specifically, using the retrieved knowledge alone (retrieved using the question as the search query), the model’s hallucination is at 38%. Using our approach of active intervention the hallucination is at 26%. We attribute our performance improvement to the active correction in the intermediate steps which eventually leads to improved answers.
> > > > > > > Similarly, in the false premise QA setting, we show this comparison in Figure 8. We note that in this case, the improvement is even larger (76% vs 24%). This is because of a recently studied concept of sycophancy, where LLMs tend to generate responses that favor the user’s perspective rather than providing correct or truthful answers, which can result in hallucinations. Our approach addresses this problem and reduces the hallucination.
> > > > > > > Advantages of our active intervention over the retrieval alone baseline:
> > > > > > > Firstly, active retrieval retrieves the knowledge that is pertinent to the current sentence in the generation. In contrast, single retrieval retrieves only once and does not have the opportunity of retrieving knowledge pertinent to the current sentence.
> > > > > > > Also, active intervention allows opportunities for course correction during the generation process i.e. if a sentence is hallucinated then it is fixed and then the subsequent sentences are generated. This prevents the propagation of hallucinations and also drives the generation in the right direction.
> > > > > > > Furthermore, single retrieval can constrain the generation to be dependent on what has been retrieved initially. In contrast, active intervention allows the model to follow its course of generation and retrieve the knowledge based on that, unlike single retrieval where the generation is based on the retrieved knowledge
> > > > > >
> > > > > > Do you have results similar to those shown in Figures 7 and 8 for the article generation task?

---

> ### Author Response · Authors · 2023-11-17
>
> Thanks for your reply. Please find our responses below:
>
> ---
>
> **1) Experiments on Latency:**
> During the rebuttal period, we have conducted an empirical analysis of the latency where we compare the latency of all the steps of the methodology.
> Figure 9 in the revised manuscript shows the comparison of latency of various steps (at a sentence level). We note that the latency of the mitigation step is low as it is only conditionally called for some sentences. We show the average mitigation latency for sentences on which it is called in the Mitigation$^*$ bar.
> We conduct this study for 10 topics (i.e., 50 sentences) for the GPT-3.5 (text-davinci-003) model.
>
> **Comparison of Overall Latency with the Generation:**
> The overall latency of the method is $2.58$ times that of the regular generation ($5354.20$ against $2071.69$).
>
> **Why the latency of the generation step is high?** This is because for the later sentences, it also takes the context in the input.
>
> **Why the latency of validation is high?**
> This is because validation procedure includes three steps (validation question creation, retrieval, and answering validation question). Furthermore, validation could be required for multiple concepts.
>
> **What does Mitigation\* represent?**
> Note that the mitigation step is only conditionally executed for some sentences. We show the average mitigation latency for sentences on which it is called in the Mitigation$^*$ bar.
>
>
> ----
>
> **5) Justification of selecting the instruction technique for the concept identification step:**
> We studied randomly sampled 50 sentences. The instruction technique identified 155 concepts in total. It missed only 2 concepts (that too these missed concepts can only be loosely regarded as important in the context of the sentence). Beyond this, please let us know what additional experiment we can perform for this step.
>
> --------
>
> **6) Is this statement is solely based on the five examples shown in Table 14?**
> No, this is an overall statement for all the instances; in Table 14, we have only shown some supporting examples.
>
> ---
>
> **7) Regarding Labeling Hallucinations:**
> Yes, such an example will be labeled as a hallucination as this is an incorrect statement. Also, this can also be regarded as an example of propagation of hallucination.
>
> ---
>
> **10)  Effectiveness of Retrieval alone:**
> We conducted this study for the multihopQA and FalsePremiseQA tasks only; in these settings, it shows a fair and clearer comparison of the efficacy of the retrieval alone method with our active intervention approach.

---

### Official Review · Reviewer_EApd · 2023-11-01

**Soundness:** 2 fair
**Presentation:** 2 fair
**Contribution:** 3 good
**Rating:** 5
**Confidence:** 4

**Summary:**

This paper proposes a method for detecting and revising hallucinations of LLMs. The proposed method takes a generated sentence as input, extracts key concepts (text spans) and identify low-confidence ones, generates a question for each span, retrieves knowledge, and revise the sentence. This is applied every time when the LLM finishes generating a sentence, so that future generations is conditioned on revised, more factual sentences. This is motived by an "propagation of hallucination" analysis , which shows that if the previous sentence is hallucinated, it is likely the next sentence generated by an LLM will also contain hallucination.

For each component (key phrase extraction, confidence estimation, query generation, retrieval), the authors empirically compared a couple variations, which suggests using prompted LLMs for all tasks, and web search for retrieval. End-to-end evaluation is done by prompting GPT 3.5 or Vicuna-13B for long article generation and manually evaluating factuality. The proposed method greatly reduces hallucination.  Additional experiments show that the proposed method can improve multi-hop QA tasks as well as identifying false-premise questions.

**Strengths:**

- The proposed method is clearly described.
- The "propagation of hallucination" analysis very nicely show the necessity of actively reducing hallucination from the generation. Although  sentence-by-sentence actively doing retrieval and rewrite has been explored in prior work, there's little quantitive analysis studying how previous hallucination can affect future generations.
- Experimental results indicate that the proposed method is very effective at reducing span-level hallucinations for long-form generation.
- The improvements on multi-hop QA is large, and the gains can be well explained by the "active" hallucination detection and revision mechanism.

**Weaknesses:**

- It would be nice to highlight the novelty of proposed framework from existing work. A very related work is [1], where the authors do active retrieval and rewrite actively when decoding each sentence, and they also use LLM output logits to find low-confidence spans for query generation. There are also several previous works that reduces LLM hallucinations at the response-level, using a similar framework as this work by prompting LLMs for span extraction, query generation, retrieval, and revise. For example, [2] and [3] uses such a framework to revise LLM responses and reduces hallucination; [4] prompted LLMs for extracting and checking claims as an automatic evaluation framework. This paper should discuss these related work, discuss the main differences, and maybe consider them as baselines.

- The paper lacks ablations to justify some of its key components. For example, though there is a strong motivation for applying the method "actively" when generating every sentence, the end-to-end evaluation does not show how it helps reduce hallucination compared to applying it at the end of the generation. Similarly, I couldn't find ablation for only fact-checking low-confidence phrases v.s. fact checking all key phrases.

- The presentation quality can be improved. Section 2 enumerates many modeling choices for each component, but it is difficult to tell what is the final method being used, and why it works better than the others. A suggestion is to describe the best approach in section 2, and leave other choices to ablation studies.  Section 3 and 4 cover many experiments, making it confusing to tell which is the most experiment and what are the main messages.

[1] Jiang, Zhengbao, et al. "Active retrieval augmented generation." arXiv preprint arXiv:2305.06983 (2023).

[2] Gao, Luyu, Zhuyun Dai, Panupong Pasupat, Anthony Chen, Arun Tejasvi Chaganty, Yicheng Fan, Vincent Zhao et al. "Rarr: Researching and revising what language models say, using language models." In Proceedings of the 61st Annual Meeting of the Association for Computational Linguistics (Volume 1: Long Papers), pp. 16477-16508. 2023.

[3] Chen, Anthony, et al. "PURR: Efficiently Editing Language Model Hallucinations by Denoising Language Model Corruptions." arXiv preprint arXiv:2305.14908 (2023).

[4] Min, Sewon, et al. "FActScore: Fine-grained Atomic Evaluation of Factual Precision in Long Form Text Generation." arXiv preprint arXiv:2305.14251 (2023).

**Questions:**

- Is there an ablation study comparing running the model actively vs running it at the end of the generation?
- Is there an ablation study comparing fact checking only the low-confidence spans vs all extracted spans?
- Since the method operates at sentence level, how does it know the context of each sentence?
- The proposed method fact-checks key phrases / named entities. I'm interested to see how it works when the hallucination is not on named entities, or when the entire sentence is made up.

---

> ### Author Response · Authors · 2023-11-16
>
> We thank the reviewer for the detailed feedback and provide our responses below:
>
> **1) Discussion on Related Work:**
> We thank the reviewer for sharing these related arxiv papers. We have included all these papers in the revised Related and Concurrent Work Section (Appendix A) and discussed the key differences. We would also like to respectfully mention that although there are key differences in our work, all these papers should be viewed as concurrent works to our paper as our work has been on arxiv within merely 1-2 months of these other arxiv papers.
>
> ---
>
> **2) Examples when the original response is completely hallucinated:**
>
> We first provide examples where the complete response is hallucinated and our approach is able to address that hallucination. Then, we discuss it quantitatively.
>
> For the topic “Ron Meagher”, the model generated “Ron Meagher is a renowned entrepreneur and business leader who has made a name for himself in the business world.” which is completely hallucinated and this hallucination also propagates in the next sentences: “He is the founder and CEO of Meagher Enterprises, a successful business consulting firm that specializes in helping companies grow and succeed.”, “Ron Meagher was born in New York City and grew up in a family of entrepreneurs.”
>
> After applying our method the result is “ Ron Meagher is an American musician best known as the bassist of the rock band The Beau Brummels. He was discovered by Autumn Records in 1964 when guitarist-songwriter Ron Elliott was putting the band together….” which is correct.
>
> Similarly, for the topic “Twila Shively”, the model generated “Twila Shively is a renowned American artist and sculptor who has been creating art for over four decades." which is completely hallucinated. In this case also, the hallucination propagates in the generation.
>
> After applying our method the result is “Twila Shively was an American competitive baseball player who played from 1945 through 1950 in the All-American Girls Professional Baseball League. She was born in Decatur, Illinois on March 20, 1922 and passed away on November 25, 1999….” which is correct.
>
> These examples demonstrate the efficacy of our approach is addressing hallucinations even when the complete sentence is made up.
> Quantitavitly, such examples fall in the “Not Related Category” in our paper and our approach addressed hallucination in 76.36% of such cases.
>
> ---
>
> **3) Actively Running our Method vs Running it at the end:**
>
> We briefly mentioned this point in Appendix B.3.3.
>
> Our detection and mitigation technique can also be applied in a ''posthoc'' manner after complete response generation. However, it has several limitations which are addressed by our ''active'' approach.
>
> The ``active'' approach prevents the propagation of hallucinations in the subsequently generated sentences, i.e., if hallucination is detected in the initially generated sentences then it would be mitigated and course correction would be done for the subsequently generated sentences. However, the ''post-hoc'' approach does not provide such an opportunity of course correction. In other words, in the ''active'' approach, the model sees the mitigated / corrected sentences while generating the subsequent sentences; thus, its output will be more correct, coherent, and fluent. In contrast, in the ''posthoc'' approach, the generated sentences are based on the initially generated previous sentences and thus the mitigated sentence will not be able to influence the generation of subsequent sentences; thus, the output would not be as coherent and fluent as the active approach.
>
> Also, applying it in a post-hoc manner will fix the sentences individually thus, redundant information could be present in multiple sentences hampering the quality of the response.
>
> For example,
>
> For the topic “Twila Shively”, the model generated “Twila Shively is a renowned American artist and sculptor who has been creating art for over four decades. She is best known for her large-scale sculptures, which often feature abstract shapes and forms. …" which is completely hallucinated.
>
> After applying our approach in a post-hoc manner gives “ Twila Shively was an American competitive baseball player who played from 1945 through 1950 in the All-American Girls Professional Baseball League. Twila Shively is known for playing baseball. …”
>
> In contrast, active approach results in “Twila Shively was an American competitive baseball player who played from 1945 through 1950 in the All-American Girls Professional Baseball League. She was born in Decatur, Illinois on March 20, 1922 and passed away on November 25, 1999. Twila began playing softball at the age of eight and quickly moved up in the softball ranks in Chicago. …"
>
> Thus, the active approach results in an output of much higher quality and doesn’t suffer from issues such are incoherence, consistency, repetition, etc.
>
> We have included these examples in Appendix B.3.3.

---

> > ### Author Response · Authors · 2023-11-16
> >
> > **4) Comparison of Fact checking keyphrases of different confidences:**
> > Figure 5 (right) shows the relationship between score and hallucination. It shows the trend of hallucination with the calculated probability score. It shows that "as the score increases, the tendency to hallucinate decreases" which implies that the model tends to hallucinate more on concepts where the score is low as compared to when the score is high.
> >
> > ---
> >
> > **5) Highlighting the Best technique for each step:**
> > To indicate the preferred technique for each step, we highlighted it in red in the Figure 1 (mentioned in the caption of the figure). We provide the justification for this preference in each step of Section 2 and further details in the Appendix. In each step of Section 3, we have highlighted (in bold) the main message of that section.
> >
> > ---
> >
> > **6) How does the model know the context while generating a sentence:**
> >
> > Please note that the previously generated sentences are used as context while generating the next sentence. So, the model indeed generates sentences in a coherent manner. This ensures that the sentences are not independent, don’t have repetition, and indeed form a coherent response.

---

> ### Comment · Reviewer_EApd · 2023-11-21
>
> Thank you for the reply!
>
> I want to further understand the following questions:
>
> **Actively Running our Method vs Running it at the end**
>
> The examples are convincing. Do you have quantitive results to demonstrate the difference between the two approaches?
>
> **Comparison of Fact checking keyphrases of different confidences**
>
> If I understand correctly, Figure 5 does not tell the end-to-end effect of fact-checking based on confidence. In other words, though there's a correlation between hallucination and confidence, it is unclear if this component affects your system's final performance.
>
> **How does the model know the context while generating a sentence**
>
> For keyphrase detection / question generation / etc, is the context being used?

---

> > ### Author Response · Authors · 2023-11-22
> >
> > Thanks for your reply. Please find our responses below:
> >
> > **Actively Running our Method vs Running it at the end:**
> >
> > Thanks for reading our response and finding our examples demonstrating the efficacy of actively running the approach convincing. We conducted this qualitative study only for this investigation as quantitative comparison of quality, repetition, coherence, etc. is challenging. We think that this kind of evaluation would go out of scope of this work. Please let us know if you would prefer to have a quantitative study on this to be included in the paper.
> >
> > **Impact of confidence score on the overall performance of the system**:
> > Thanks for this suggestion. During the rebuttal phase, we have conducted a study comparing the overall hallucination percentage for difference score values.
> >
> > | score | Hallucination % |
> > |--------------|--------------------|
> > 0.25 | 68 |
> > 0.35 | 36 |
> > 0.55 | 20 |
> > 0.75 | 16 |
> > 0.85 | 16 |
> >
> > The above study is conducted on 5 topics (i.e. 25 sentences). This result follows the following trend: As the score increases the hallucination decreases. This is expected because a lower score means that the system considers very few concepts as uncertain and thus lets them pass without validation leading to higher hallucination. In contrast, when the score is high, many concepts get validated leading to lower hallucination.
> >
> > **For keyphrase detection / question generation / etc, is the context being used?**
> > The context (previously generated sentences) is not used in these steps. However, for the article generation task, we provide the topic name in the prompt as shown in Table 3.

---

### Official Review · Reviewer_5fdS · 2023-11-04

**Soundness:** 2 fair
**Presentation:** 3 good
**Contribution:** 3 good
**Rating:** 5
**Confidence:** 3

**Summary:**

This paper presents a method to reduce LLM hallucinations using an early detection approach. The paper demonstrates that active detection and mitigation of hallucinations using logit output values is a viable path. The paper presents results on GPT 3.5 and Vicuna.

**Strengths:**

The paper works on an important problem of mitigating hallucinations
The paper presents an early detection approach and demonstrates effectiveness with two LLMs
The paper is extremely well-written, with clear goals, a well-described approach, and a detailed Appendix

While it is always possible to nitpick on experimental design issues, we need to be mindful of the fact that this work is presented within the scope of a single ICLR submission. With that in mind, the paper does an excellent job.

**Weaknesses:**

It is unclear how effective this method will be for generations beyond the first five sentences.

Post-rebuttal response: After going over the discussions, reviews, and rebuttals. I feel that my initial assessment of the paper had gaps. I lean towards the majority view that the paper needs improvement. I have updated my scores accordingly.

**Questions:**

It is unclear how effective this method will be for generations beyond the first five sentences. Will it be more useful to distribute these checkpoints across the generated text?

---

> ### Author Response · Authors · 2023-11-14
>
> We thank the reviewer for the encouraging feedback. Please find our responses to your questions below:
>
> --------
>
> **Discussion on the effectiveness of the method beyond the first five generated sentences:**
>
> Thanks for this thoughtful question. Our study on the article generation task is conducted on the first five generated sentences. After applying our method, the correctness at sentence number level (averaged over all the inputs) is as follows (Sentence 1: 90.0%, Sentence 2: 82.67%, Sentence 3: 86.67%, Sentence 4: 82.67%, Sentence 5: 85.34%). These values are indeed close and do not considerably reduce as the sentence number increases. With this result, we show that our method of active detection and mitigation successfully mitigates the hallucination throughout the generation (not restricted to any specific sentence number). Furthermore, it shows that the ability to address hallucinations does not considerably diminish as the sentence number increases. Thus, even increasing the number of sentences is not expected to considerably impact the improvement that our method would bring. We will also include this additional result with the corresponding discussion in the revised version.
>
> --------
>
> **Will it be more useful to distribute these checkpoints across the generated text?**
>
> Our interpretation of this question is "why the unit of generation is a sentence and not multiple sentences."
>
> We select a unit as a sentence over multiple sentences and (also over just a few words instead of a sentence) because of the following reasons.
>
> Why not multiple sentences?
>
> In autoregressive generation, the generation depends on the context including the model’s previously generated text. Thus, if we consider multiple sentences as a unit in our approach (let’s say 3 sentences) and if one of the initial sentences is hallucinated (and thus replaced with the corrected sentence), the subsequent sentences (i.e., the remaining sentences of the unit) may not stand relevant (as they were based on a sentence that has been replaced) and it may make the generation incoherent. Furthermore, the propagation of hallucination is another negative contributor as the next sentences may carry forward the hallucination of the previous incorrect sentences. Thus, the subsequent sentences in the unit would need to be regenerated. This implies that using multiple sentences as a unit may not return that benefit (that too at the extra cost of generating multiple sentences at once).
>
> Similarly, we justify why we do not use a few words as a unit instead of a sentence.
>
> We note that using a few words (i.e., a window of text) may not have sufficient information to test the correctness of the concepts in the generation. For instance, if the window is of the following words: “Rick Mahler won three gold medals and 2 silver medals at the”, it doesn’t have sufficient information to validate the correctness of the individual concepts. On the other hand, a sentence typically provides richer context to validate the correctness of the concepts of the sentence.
>
> Because of the above two reasons, we use a sentence as the unit in our method.

---

### Official Review · Reviewer_pdV3 · 2023-11-08

**Soundness:** 1 poor
**Presentation:** 2 fair
**Contribution:** 1 poor
**Rating:** 3
**Confidence:** 4

**Summary:**

The paper proposes a method for detecting and mitigating hallucinations in LLM outputs. The detection consists of finding "important" concepts in the output, filtering them based on the model uncertainty, conducting web-search and feeding the info to the model to answer if the output contained hallucinations. The paper also proposes to use this knowledge from web-search to mitigate hallucinations.

**Strengths:**

The paper addresses the topic of hallucinations which is a relevant and timely topic.

**Weaknesses:**

1. The paper does not mentioned the highly relevant work of Kadavath et al., [Language Models (Mostly) Know What They Know](https://arxiv.org/abs/2207.05221), which also uses the model uncertainty to detect hallucinations. Since uncertainty is used as a major signal in the proposed pipeline, the novelty of the proposed approach is not clear.

2. The paper does not study important choices in details. For instance, the web search procedure is not very clear. The paper says “In some cases, multiple web searches were required to check the correctness of different facets of a sentence”. Are these searched human-supervised? What are the stopping criteria? I would suggest adding the web-search procedure in an algorithm block so that the readers can understand it better.

3. Similarly, the paper does not discuss exactly what kind of "important" concepts are identified by the model. Could you provide some examples? Are the models supposed to extract all relevant concepts? Is the concept extraction supposed to work well across different application domains (e.g., questions answering)? What if we are working with non-instruction tuned models?

4. It is not clear how good the instruction models were at following different instructions in Table 3. Did the authors perform a systematic analysis here?

**Questions:**

See points 1-4 under "Weaknesses".

---

> ### Author Response · Authors · 2023-11-14
>
> We thank the reviewer for the comments. Please see our responses to your comments below.
>
> ------------
>
> **1) Distinguishing our work from the paper “Language Models (Mostly) Know What They Know”**: In our revised manuscript, we will include a citation to this paper and provide more discussion in the expanded related work (Appendix A). While we acknowledge that this paper is related to our research, we respectfully disagree with the reviewer’s comment that not citing Kadavath’s work is a major weakness. The focus and scope of Kadavath’s work are different from ours.
>
> The paper “Language Models (Mostly) Know What They Know” has the following major findings:
> - larger models are well-calibrated on multiple-choice and true/false questions
> - Calibration improves with model size, and it also improves when we pass from zero-shot to few-shot evaluation;
> - Explored self-evaluation, i.e., whether the output generated by the model is True or False. Section 4.1 (page in this paper) shows an example of  Basic Self-Evaluation.
>
> In contrast, our paper proposed an end-to-end methodology to detect and mitigate the hallucinations in the output of the LLMs. Specifically, we start by identifying the candidates of potential hallucination leveraging the model’s logit output values, check their correctness through a validation procedure (by retrieving the pertinent knowledge), mitigate the detected hallucinations via prompting, and then continue with the generation process. We show that this active intervention also helps in preventing the propagation of hallucination in the model’s autoregressively generated output.
>
> We note that the focus and scope of our work are different from Kadavath’s work as the only common string is the use of logit values (which is just a component in our end-to-end methodology of detecting and mitigating the hallucinations in the output of the LLMs). We explain the differences in the use of logit values in more detail below:
>
> We calculate the “concept-level” uncertainty of the model in “open-ended” long text generation using our techniques (presented in section 2.1.2  in our paper), namely, Average of Probabilities, Normalized Product of Probabilities, and Minimum of Probabilities unlike Kadavath’s work which focuses on "response-level" and evaluates calibration on multiple choice, true/false questions, and tasks like TriviaQA which are non-trivially different from open-ended article generation. Also, the “Self-Inquiry” technique in our work (which is although not a part of our main methodology as we use web search as the preferred technique for the retrieval step) is at the concept level and automatically creates validation inquiry questions based on the sentence and the concept (Section 2.1.3). In contrast, Kadavath’s paper used the following template: “Is the proposed answer:(A) True (B) False. The proposed answer is:” which is not suitable for open-ended long generation which typically has multiple facets in a single sentence.
>
> Furthermore, we additionally use our proposed validation procedure (Section 2.1.3-2.1.5) to check the correctness of the uncertain concepts rather than just relying on the uncertainty values to detect hallucinations.
>
> In summary, the objective, the focus, and the scope of our work are different from Kadavath’s work. The way we use the logit output values, the way we identify the concepts, the way we calculate the score corresponding to the identified concepts and the way we validate the correctness of the concept are different from Kadavath’s work.

---

> ### Author Response · Authors · 2023-11-14
>
> **2) Clarification on web search procedure for annotation:** We note that the line “In some cases, multiple web searches were required to check the correctness of different facets of a sentence” in our paper in Section 3 is written in the context of the collection of human annotations for the hallucination task (and this is not a part of the methodology).
> As specified in Section 3, the human annotation for this task requires validating the correctness of the generation. Since the generation is for a variety of topics of different domains and would be beyond the common knowledge of a typical human, thus, we use web search to gather the relevant information to check the correctness of the generation. Multiple web searches were required in some cases because a generation can contain multiple facets of information all of which can not be validated in a single web search.
>
> For example, sentences like “​Steven Threet i​s best known for his time at the University of Michigan, where he was a three-year starter and led the Wolverines to a Big Ten Championship in 2008.”, "Rick Mahler was a Major League Baseball pitcher who played for the Atlanta Braves, Cincinnati Reds, and St. Louis Cardinals from 1979 to 1994.” contains multiple facets that need to be validated separately because a single web search may not return all the information that is necessary to validate the correctness of all the facets of such sentences.
>
> In the proposed active detection and mitigation method, we use the standard web search procedure (as detailed in 2.1.4), i.e, the validation query created in the previous step is used as the search query. We use Bing search API for retrieving the pertinent knowledge. Using a query to obtain the search results is a widely used standard procedure.
>
> -----------
>
> **3) Further Explanation on the concept identification step**:
> Firstly, we underline the importance of the concept identification step. It is crucial because validating the correctness of the entire sentence at once is infeasible as a sentence often contains multiple different facets all of which can not be validated at once. In contrast, individually validating correctness corresponding to the concepts provides opportunities for accurately detecting incorrectness.
>
> We define “important” concepts as those that can be potential hallucinations in the generation and thus need to be validated. We provide the motivation behind this step in Section 2.1.1.
>
> We note that the large language models are considerably good at understanding such instructions and the success of a variety of prompting-oriented methods (including chain of thought, Least to Most, etc) is a testament to that.
> Though this task is considerably straight-forward for LLMs (even for those of relatively smaller sizes), in cases where a specific LLM under study is not capable of following the instructions, we have provided alternative techniques for this step, namely Entity Extraction and Keyword Extraction which do not rely on the LLM and instead use an external tool for this step.
>
> Please refer to Table 4 and Table 5 in the Appendix where we provide examples of the identified concepts in this step.
>
> -----------
>
> **4) Performance of models in following different instructions presented in Table 3:**
> Yes, we have indeed conducted evaluations of the efficacy of the instructions. Specifically, for the concept identification step, we studied randomly sampled 50 sentences. The instruction technique identified 155 concepts in total. It missed only 2 concepts (that too these missed concepts can only be loosely regarded as important in the context of the sentence). We have only provided a qualitative study for this step in the Appendix and will include this in the revision. Furthermore, the efficacy of the validation and mitigation instructions is presented in Table 1 and 2, respectively.
>
> We note that the overall efficacy of these techniques (and how well they serve their purpose) is evaluated by the overall improvement in reducing the hallucinations.
>
> We also note that the LLM can be prompted in a different way also to achieve the same objective; however, the purpose of this work is to show that the complex task of addressing hallucinations in an end-to-end manner can be decomposed into simpler steps that can be solved via instructing the model.

---

> ### Comment · Area_Chair_CP2y · 2023-11-22
>
> Dear reviewer pdV3, given the authors detailed response, please provide your updated feedback.  Even if your score/decision does not change, it is critical to engage in the discussion.  Thank you for your important contribution to this conference.

---

### Author Response · Authors · 2023-11-22
**Thanking for the Insightful Reviews + Revision Summary**

We sincerely thank all the reviewers for their insightful feedback and appreciate their hard work in reviewing our paper.

We are encouraged and motivated that the reviewers find our research problem on mitigating LLM hallucinations to be important and timely and our approach to be clear and effective.

We answer the reviewer's questions in the individual responses.

We have revised the paper incorporating the comments and suggestions. Below is the summary of the revisions:

1. A discussion on latency of the method (B.6.1) - As mentioned by the reviewer that though the latency may not be a problem for all use cases, it would be important to have a detailed discussion on it. We have included a detailed discussion on it for all the steps of the approach with suggestions on their lower-cost alternatives. We have also included an empirical analysis of the latency where we compare the latency of all the steps of the methodology (Figure 9).

2. Including suggested related and concurrent work - We have included all the suggested related and concurrent papers in the revision.

3.  Discussion on effectiveness of the method beyond the five sentences - We have included this discussion in Appendix J.

4. Justification for using a sentence as the unit of generation in the method - We have included this justification in the Design Decisions Section (Appendix B.4).

5. Including Examples of various scenarios - We have included examples of scenarios such as when the original response is completely hallucinated and post-hoc variant of the approach.

6. Impact of the probability score on overall performance - We conducted this experiment and the results showed that as the probability score is increased, the hallucination percentage reduces as many concepts get validated.

7. Presentation-related changes - We have incorporated the suggestions in the revision.